# Local circuit allowing hypothalamic control of hippocampal area CA2 activity and consequences for CA1

Vincent Robert[1], Ludivine Therreau[1], Vivien Chevaleyre[1,2], Eude Lepicard[1], Cécile Viollet[1], Julie Cognet[1], Arthur JY Huang[3], Roman Boehringer[3], Denis Polygalov[3], Thomas J McHugh[3], Rebecca Ann Piskorowski[1,2]*

[1]INSERM UMR1266, Institute of Psychiatry and Neuroscience of Paris, Team Synaptic Plasticity and Neural Networks, Université de Paris, Paris, France; [2]GHU Paris Psychiatrie and Neurosciences, Paris, France; [3]Laboratory for Circuit and Behavioral Physiology, RIKEN Center for Brain Science, Saitama, Japan

**Abstract** The hippocampus is critical for memory formation. The hypothalamic supramammillary nucleus (SuM) sends long-range projections to hippocampal area CA2. While the SuM-CA2 connection is critical for social memory, how this input acts on the local circuit is unknown. Using transgenic mice, we found that SuM axon stimulation elicited mixed excitatory and inhibitory responses in area CA2 pyramidal neurons (PNs). Parvalbumin-expressing basket cells were largely responsible for the feedforward inhibitory drive of SuM over area CA2. Inhibition recruited by the SuM input onto CA2 PNs increased the precision of action potential firing both in conditions of low and high cholinergic tone. Furthermore, SuM stimulation in area CA2 modulated CA1 activity, indicating that synchronized CA2 output drives a pulsed inhibition in area CA1. Hence, the network revealed here lays basis for understanding how SuM activity directly acts on the local hippocampal circuit to allow social memory encoding.

*For correspondence:
rebecca.piskorowski@inserm.fr

**Competing interests:** The authors declare that no competing interests exist.

## Introduction

The hippocampus is critical for memory formation and spatial navigation (*Buzsáki and Moser, 2013*; *Eichenbaum and Cohen, 2014*), yet basic questions persist regarding the underlying circuitry and cellular components. While area CA2 has been shown to play a significant role in several hippocampal processes including social memory formation (*Hitti and Siegelbaum, 2014*; *Stevenson and Caldwell, 2014*) sharp-wave ripple generation (*Oliva et al., 2016a*) and spatial encoding (*Kay et al., 2016*), information about the local circuitry and cellular mechanisms allowing these functions is lacking. There is mounting evidence that generalizations cannot be made from the rich understanding of areas CA1 and CA3, as neurons in area CA2 have been shown to have unique molecular expression profiles (*Cembrowski et al., 2016*; *Lein et al., 2004*), morphology (*Bartesaghi and Ravasi, 1999*; *Nò, 1934*) and cellular properties (*Robert et al., 2020*; *Srinivas et al., 2017*; *Sun et al., 2014*). Notably, and in contrast to area CA1, CA2 pyramidal neurons do not undergo high-frequency stimulation-induced synaptic plasticity (*Dasgupta et al., 2020*; *Zhao et al., 2007*). Rather, the excitability of this region is tightly controlled by a highly plastic network of inhibitory neurons (*Leroy et al., 2017*; *Nasrallah et al., 2015*; *Piskorowski and Chevaleyre, 2013*). When active, CA2 pyramidal neurons (PNs) can strongly drive area CA1 (*Chevaleyre and Siegelbaum, 2010*; *Kohara et al., 2014*; *Nasrallah et al., 2019*), thereby influencing hippocampal output. Furthermore, CA2 neurons also project to area CA3, where they recruit inhibition (*Boehringer et al., 2017*; *Kohara et al., 2014*) and act to control hippocampal excitability. Thus, CA2 neurons are poised to have long-

reaching effects in the hippocampus, and a better understanding of the regulation of neuronal activity in this region is needed.

The hypothalamic supramammillary (SuM) nucleus sends projections to both area CA2 and the dentate gyrus (DG) (*Haglund et al., 1984*; *Vertes, 1992*). These long-range connections have been shown in several species including rodents, primates, and humans (*Berger et al., 2001*; *Haglund et al., 1984*; *Wyss et al., 1979*) where they are present in early hippocampal development. The SuM has been found to be active during a wide variety of conditions including novel environment exposure (*Ito et al., 2009*), reinforcement learning (*Ikemoto, 2005*; *Ikemoto et al., 2004*), food anticipation (*Le May et al., 2019*), and during REM sleep and arousal (*Pedersen et al., 2017*; *Renouard et al., 2015*). This nucleus is also known for participating in hippocampal theta rhythm (*Pan and McNaughton, 2002*; *Pan and McNaughton, 1997*), possibly by its direct projection to the hippocampus or by modulation of the medial septum (*Borhegyi et al., 1998*; *Vertes and Kocsis, 1997*), and regulating spike-timing between hippocampus and the cortex (*Ito et al., 2018*). Disruption of SuM neuron activity with pharmacological methods (*Aranda et al., 2008*; *Shahidi et al., 2004*) or lesions (*Aranda et al., 2006*) has been reported to disrupt hippocampal memory. Serotonin depletion of the SuM leads to deficiencies in spatial learning in the Morris water maze, and results in altered hippocampal theta activity (*Gutiérrez-Guzmán et al., 2012*; *Hernández-Pérez et al., 2015*). Salient rewarding experiences also activate the SuM, as evidenced by cFos expression in monoaminergic SuM neurons by consumption of rewarding food (*Plaisier et al., 2020*). Furthermore, the rewarding aspects of social aggression have been shown to involve an excitatory circuit between the hypothalamic ventral premammillary nucleus and the SuM (*Stagkourakis et al., 2018*). It has recently been shown that there are two separate populations of cells in the SuM that target either CA2 or the DG (*Chen et al., 2020*). In the DG, the SuM terminals release both glutamate and GABA (*Boulland et al., 2009*; *Chen et al., 2020*; *Hashimotodani et al., 2018*; *Pedersen et al., 2017*; *Soussi et al., 2010*). The SuM-DG projection has been recently shown to play a role in modulating DG activity in response to contextual novelty (*Chen et al., 2020*) and spatial memory retrieval (*Li et al., 2020*). In contrast, functional studies of the SuM-CA2 projection have found that this connection is entirely glutamatergic (*Chen et al., 2020*). It was recently discovered that the CA2-projecting SuM neurons are active during social novelty exposure, and their selective stimulation prevents expression of a memory of a familiar conspecific (*Chen et al., 2020*). These findings strongly suggest that the SuM-CA2 connection conveys a social novelty signal to the hippocampus. Furthermore, recent in vivo recordings from the SuM in anesthetized rats reported that a subset of SuM neurons were active earlier than CA2 and other hippocampal cells during SWR (*Vicente et al., 2020*), indicating a possible role for the SuM-CA2 projection in shaping area CA2 activity prior to SWR onset.

Even with the anatomical and in vivo data, the properties and consequences of SuM activation on area CA2 activity remain unexplored. In this study, we use a combination of approaches to specifically examine the effects of SuM input stimulation on neuronal activity in hippocampal area CA2. Here, we show that the SuM-evoked post-synaptic excitation of CA2 PN is controlled by SuM-driven inhibition. We identified PV-expressing basket cells as the neuronal population most strongly excited by SuM input in area CA2, and thus likely responsible for the feedforward inhibition evoked by SuM in CA2 PNs. We found that recruitment of this inhibition enhances the precision of AP firing by area CA2 PNs in conditions of low and high cholinergic tone. Finally, we observed that the resulting synchronized CA2 PN activity drives inhibition in area CA1, thereby providing a circuit mechanism through which SuM can modulate hippocampal excitability by controlling area CA2 output.

## Results

### SuM axons provide excitatory glutamatergic input to pyramidal neurons in area CA2 and CA3a

Its small size and cellular heterogeneity have made the SuM a difficult region to study. It has been shown that the source of vesicular glutamate transporter 2 (VGluT2)-immunopositive boutons in area CA2 originate from the SuM (*Halasy et al., 2004*). In order to more closely examine the SuM-CA2 long-range connection, we injected a retrograde canine adenovirus type 2 (CAV-2) into area CA2 of the hippocampus to permit the expression of Cre-recombinase (Cre) in hippocampal-projecting SuM neurons, and an adeno-associated virus (AAV) was injected into the SuM to allow the expression of

EGFP under the control of Cre (*Figure 1—figure supplement 1A*). In five animals the injection of retrograde CAV-2 was sufficiently targeted to area CA2, as indicated by the presence of EGFP-expressing SuM axonal fibers primarily in this hippocampal area (*Figure 1—figure supplement 1B*). We stained for calretinin to define the boundaries of the SuM nucleus (*Pan and McNaughton, 2004*). Consistent with recent findings using retrograde AAV vectors (*Chen et al., 2020*), we observed that CA2-projecting cells express calretinin and are located in the medial SuM (*Figure 1—figure supplement 1C–D*). These cells were located bilaterally, ventral to the fiber bundles that traverse the SuM (*Figure 1—figure supplement 1C*). Furthermore, we confirmed that these cells also stain for VGluT2 (*Figure 1—figure supplement 1E*).

In order to better understand the cellular targets and consequences of SuM input activity in area CA2, we injected an AAV to express channelrhodopsin(H143R)-YFP (ChR2-EYFP) under the control of Cre into the SuM of a transgenic mouse line with Cre expression controlled by the VGluT2 promoter, the *Tg(Slc17a-icre)10Ki* line (*Borgius et al., 2010*; *Figure 1—figure supplement 1F*). In parallel, we used the *Csf2rb2-Cre* mouse line that selectively expresses *Cre* in the SuM (*Chen et al., 2020*; *Figure 1A*). We found that with both transgenic mouse lines we could reproducibly restrict expression of ChR2-EYFP in the SuM and avoid infecting nearby hypothalamic regions that also project to the hippocampus (*Figure 1A*, *Figure 1—figure supplement 1F*). Furthermore, with both lines of transgenic mice, we observed identical patterns of SuM fiber localization in the hippocampus. EYFP-containing SuM axons were found throughout the granule cell layer of the DG and in area CA2 (*Figure 1B*) where they clustered around the pyramidal layer (*stratum pyramidale*, SP). The SuM fiber projection area was clearly restricted to area CA2, as defined by expression of the CA2-specific markers PCP4 (*Figure 1—figure supplement 1B*) and RGS14 (*Figure 1B*), and did not spread to neighboring areas CA3 and CA1. In order to maximize the precision of our experiments, we frequently only achieved partial infection of the SuM, as indicated by the sparseness of ChR2-EYFP-containing fibers in comparison to the number of VGluT2-stained boutons in this region (*Figure 1—figure supplement 1G–H*).

We performed whole-cell current and voltage clamp recordings of PNs across the hippocampal CA regions and activated projecting axons with pulses of 488 nm light in acute hippocampal slices. Following all recordings, we performed post-hoc anatomical reconstructions of recorded cells and axonal fibers, as well as immunohistochemical staining for CA2-area markers. Additionally, injection sites were examined post hoc to ensure correct targeting of the SuM.

We observed that photostimulation of SuM axons elicited excitatory post-synaptic responses in 63% of PNs (n = 166 of 263 cells) located in area CA2. PNs in this region shared similar overall dendritic morphologies and electrophysiological properties (*Table 1*) but differed along two criteria. First, in *stratum lucidum* where the DG mossy fibers (MF) project, some PNs clearly had thorny excrescences (TE) while others had very smooth apical dendrites (*Figure 1C–D*). Based on the presence of TEs, we classified cells as CA2 or CA3a PNs (unequivocal distinction was possible for 148 neurons). Second, the distribution of the locations of PN soma along the radial axis of the hippocampus allowed us to cluster them as deep (closer to *stratum oriens*, SO) or superficial (closer to *stratum radiatum*, SR) subpopulations (unequivocal distinction was possible for 157 neurons). We found that the SuM-PN connectivity was not different between CA2 and CA3 PNs (*Table 2*, $\chi^2$ test for CA2 and CA3 PNs, p=0.572) or between deep and superficial PNs (*Table 2*, $\chi^2$ test for deep and superficial PNs, p=0.946). Light-evoked excitatory post-synaptic potentials (EPSPs) and excitatory post-synaptic currents (EPSCs) recorded at −70 mV were of fairly small amplitude (*Figure 1C–D*) that were similar regardless of the PN type or somatic location (*Table 2*, Mann-Whitney U test for CA2 and CA3 PNs, p=0.409; Mann-Whitney U test for deep and superficial PNs, p=0.306). Because no significant differences in post-synaptic responses to SuM input stimulation were observed between CA2 and CA3 PNs as well as between deep and superficial PNs, data from all PNs was pooled for the rest of the study. The small amplitude of SuM input-evoked post-synaptic responses in PNs was not due to suboptimal stimulation of SuM axons as EPSC amplitudes rapidly reached a plateau when increasing light intensity (*Figure 1—figure supplement 2A–B*). We are confident that this transmission is due to action potential-generated vesicle release because all transmission was blocked following application of the sodium channel blocker tetrodotoxin (TTX) (*Figure 1—figure supplement 2B*). The pure glutamatergic nature of the SuM input was confirmed by the complete block of light-evoked synaptic transmission following the application of the AMPA and NMDA receptors antagonists NBQX and D-APV (*Figure 1—figure supplement 2C*; amplitudes were −16 ± 4.8 pA in control and −1.8 ± 0.3

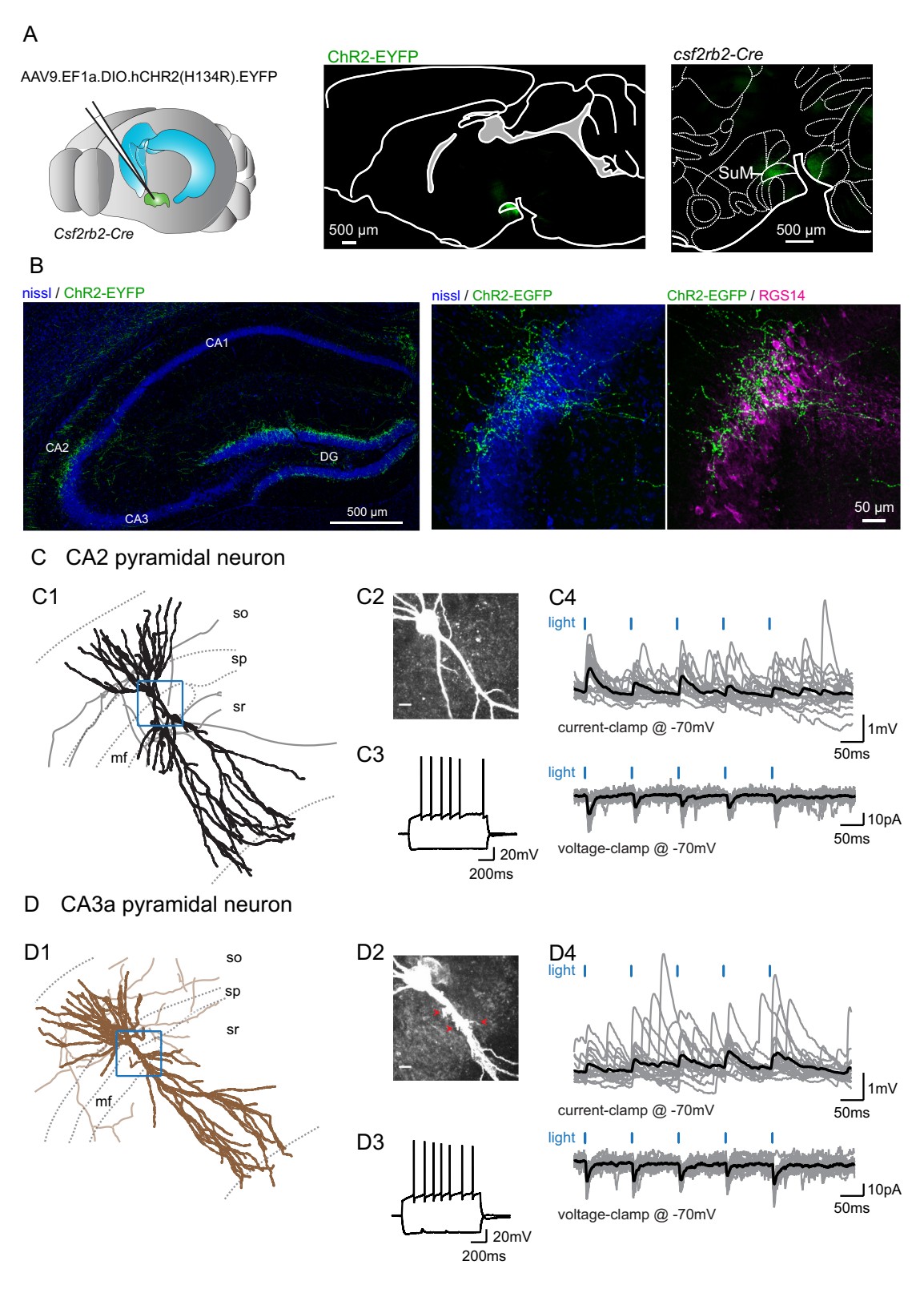

**Figure 1.** Selective functional mapping of SuM neurons that project to hippocampal area CA2. (**A**) Left, diagram illustrating the injection of AAVs into the SuM. Middle, sagittal image indicating the infected SuM area expressing hCHR2(H134R)-EYFP (green). Right, expanded view of injection site in the Csf2rbr-Cre mouse line. (**B**) Left, hCHR2(H134R)-EYFP -expressing SuM fibers (green) and nissl staining (blue) in the hippocampus. Right, higher magnification image of area CA2 with hCHR2(H134R)-EYFP -expressing SuM fibers (green) and nissl staining (blue) and RGS14 staining (magenta) to

*Figure 1 continued on next page*

*Figure 1 continued*

label area CA2. (**C**) CA2 pyramidal neurons in the SuM-innervated region receive excitatory transmission. (C1) Example CA2 PN reconstruction (dendrites in black, axons in gray, hippocampal stratum borders shown in dotted line, area demarcated in blue corresponds to the expanded image in C2). (C2) Biocytin labeling of the recorded cell proximal dendrites, scale bar represents 10 μm. (C3) AP firing and repolarizing sag potential in response to steps of +800 and −400 pA current injection. (C4) Light-evoked EPSPs (top traces, individual traces shown in gray, average trace shown in black) and EPSCs (bottom traces, individual traces shown in gray, average trace shown in black). (**D**) CA3a pyramidal neurons in the SuM-innervated region receive excitatory transmission. (D1) Example CA3 PN reconstruction (dendrites in brown, axons in light brown, hippocampal stratum borders shown in dotted line, area demarcated in blue corresponds to the expanded image in D2). (D2) Biocytin labeling of the recorded cell proximal dendrites, note the presence of thorny excrescences, as indicated by the red arrows; scale bar represents 10 μm. (D3) AP firing and repolarizing sag potential in response to steps of +800 and −400 pA current injection. (D4) Light-evoked EPSPs (top traces, individual traces shown in gray, average trace shown in black) and EPSCs (bottom traces, individual traces shown in gray, average trace shown in black).

The online version of this article includes the following figure supplement(s) for figure 1:

**Figure supplement 1.** Anatomy and immunohistochemistry of hippocampal-projecting SuM neurons.
**Figure supplement 2.** Glutamate is the only neurotransmitter at the SuM-CA2 synapse.

pA in NBQX and D-APV, n = 6; Wilcoxon signed-rank test, p=0.03). These data confirm that SuM inputs provide long-range glutamatergic excitation to CA2 and CA3 PNs in area CA2.

## PNs in area CA2 receive mixed excitatory and inhibitory responses from the SuM input

Using whole-cell voltage clamp recordings in area CA2 and the dentate gyrus (DG), we have previously shown that the CA2-targeting and DG-targeting SuM neurons have contrasting neurotransmitter modalities (*Chen et al., 2020*). Our results and other have demonstrated that glutamate and GABA are co-released at SuM-DG synapses (*Boulland et al., 2009*; *Chen et al., 2020*; *Hashimotodani et al., 2018*; *Pedersen et al., 2017*; *Soussi et al., 2010*), but that the SuM-CA2 synapses are exclusively glutamatergic (*Chen et al., 2020*). We have previously shown that SuM input stimulation in area CA2 evokes a very large inhibitory post-synaptic current (IPSC) that is entirely due to feedforward inhibition based on the delayed response latencies of IPSCs as compared to EPSCs, the complete block of IPSCs by NBQX and APV, and the complete abolition of IPSCs but sparing of EPSCs in the presence of TTX and 4-amino pyridine (*Chen et al., 2020*). Because photostimulation of SuM input elicited excitatory post-synaptic potentials (PSPs) of fairly small amplitude in area CA2 PNs held at −70 mV (*Figure 1C* 4 and D4), we asked if the amplitude of SuM input stimulation-evoked PSPs in PNs could be controlled by feedforward inhibition. Interestingly, blocking inhibitory transmission with the GABA$_A$ and GABA$_B$ receptor antagonists SR95531 and CGP55845A led to a significant increase of light-evoked PSP amplitude recorded in area CA2 PNs (*Figure 2A–C*; amplitudes of the first response were 0.18 ± 0.05 mV in control and 0.24 ± 0.05 mV in SR95531 and CGP55845A, n = 14; Wilcoxon signed-rank tests, p=0.004 for the first PSP, p=0.013 for the second PSP, p<0.001 for the third PSP). Thus, this result demonstrates a negative control of SuM-driven excitation by feedforward inhibition.

Given the combination of direct excitation and feedforward inhibition from SuM inputs onto CA2 pyramidal cells, we asked how this input would summate with other synaptic inputs in the CA2 dendritic arbor. Hippocampal area CA2 receives synaptic input from CA3 in *stratum radiatum* (SR).

**Table 1.** Electrophysiological properties of pyramidal neurons in SuM-innervated area.

| | V$_M$ (mV) | R$_M$ (MOhm) | C$_M$ (pF) |
|---|---|---|---|
| CA2 PN (n = 81) | −69.8 ± 0.70 | 59.2 ± 2.65 | 209 ± 11.4 |
| CA3 PN (n = 31) | −70.3 ± 1.06 | 72.4 ± 4.82 | 211 ± 15.7 |
| Statistics | Mann-Whitney U test p=0.997 | Student T test p=0.020* | Mann-Whitney U test p=0.625 |
| PN deep (n = 57) | −71.1 ± 0.76 | 64.0 ± 3.94 | 200 ± 12.3 |
| PN superficial (n = 76) | −69.3 ± 0.67 | 64.9 ± 3.19 | 196 ± 11.8 |
| Statistics | Student T test p=0.077 | Mann-Whitney U test p=0.777 | Mann-Whitney U test p=0.588 |

**Table 2.** Characteristics of SuM light-evoked transmission onto pyramidal neurons.

| Cell type | EPSC | | | | | |
|---|---|---|---|---|---|---|
| | Connectivity (%) | Amplitude (pA) | Rise time (ms) | Decay time (ms) | Latency (ms) | Success rate |
| CA2 PN | 56 (n = 58 of 103) | −16 ± 1.9 | 2.9 ± 0.1 | 14 ± 0.8 | 2.4 ± 0.2 | 0.44 ± 0.03 |
| CA3 PN | 49 (n = 22 of 45) | −23 ± 5.9 | 3.0 ± 0.2 | 14 ± 0.9 | 2.7 ± 0.3 | 0.56 ± 0.06 |
| Statistics | $\chi^2$ test p=0.572 | Mann-Whitney U test p=0.409 | Mann-Whitney U test p=0.391 | Mann-Whitney U test p=0.797 | Mann-Whitney U test p=0.156 | Student T test p=0.074 |
| PN deep | 56 (n = 35 of 63) | −15 ± 2.0 | 3.5 ± 0.2 | 16 ± 1.0 | 3.5 ± 0.4 | 0.39 ± 0.03 |
| PN superficial | 56 (n = 53 of 94) | −20 ± 3.0 | 3.1 ± 0.2 | 15 ± 0.9 | 2.7 ± 0.3 | 0.51 ± 0.04 |
| Statistics | $\chi^2$ test p=0.946 | Mann-Whitney U test p=0.306 | Mann-Whitney U test p=0.051 | Mann-Whitney U test p=0.314 | Mann-Whitney U test p=0.083 | Mann-Whitney U test p=0.072 |
| Cell type | IPSC | | | | | |
| | Connectivity (%) | Amplitude (pA) | Rise time (ms) | Decay time (ms) | Latency (ms) | Success rate |
| CA2 PN | 35 (n = 19 of 55) | 197 ± 41.3 | 3.8 ± 0.4 | 25 ± 1.2 | 6.3 ± 0.7 | 0.55 ± 0.06 |
| CA3 PN | 57 (n = 16 of 28) | 145 ± 23.4 | 4.5 ± 0.4 | 25 ± 1.2 | 7.5 ± 0.9 | 0.54 ± 0.05 |
| Statistics | $\chi^2$ test p=0.134 | Mann-Whitney U test p=0.870 | Student T test p=0.203 | Mann-Whitney U test p=0.896 | Mann-Whitney U test p=0.303 | Student T test p=0.893 |
| PN deep | 47 (n = 16 of 34) | 199 ± 40.6 | 3.8 ± 0.4 | 25 ± 1.4 | 7.2 ± 0.8 | 0.52 ± 0.07 |
| PN superficial | 47 (n = 26 of 55) | 167 ± 27.5 | 4.9 ± 0.4 | 26 ± 1.2 | 6.8 ± 0.7 | 0.50 ± 0.05 |
| Statistics | $\chi^2$ test p=0.987 | Mann-Whitney U test p=0.258 | Student T test p=0.047* | Student T test p=0.564 | Student T test p=0.706 | Student T test p=0.796 |

Stimulation of CA3 inputs evokes a very strong feedforward inhibition, such that it is exceptionally difficult to evoke action potential firing in CA2 pyramidal neurons when inhibitory transmission is intact (*Chevaleyre and Siegelbaum, 2010*; *Nasrallah et al., 2015*; *Piskorowski and Chevaleyre, 2013*). Additionally, CA2 PNs receive synaptic input from the entorhinal cortex in *stratum lacunosum molecular* (SLM). These inputs are very distal but relatively less attenuated in CA2 PNs in comparison to distal inputs in CA1 (*Chevaleyre and Siegelbaum, 2010*; *Srinivas et al., 2017*). In order to answer how the SuM input interacts with the CA3 and entorhinal inputs in area CA2, we electrically stimulated synaptic inputs in SR and SLM in the presence and absence of simultaneous SuM fiber stimulation (*Figure 2D*). In summary, we found that when the CA2 PNs were kept at −70 mV, SuM input stimulation paired with SR or SLM input had a net depolarizing effect. We measured the amplitudes of the light-evoked SuM PSP, the electrically evoked PSP of either SR or SLM stimulation and the paired SuM and electrical PSP (*Figure 2E*). For SR input stimulation, we found no significant difference between the observed paired SR+SuM amplitude and the calculated linear summated amplitude (SR alone + SuM alone) (*Figure 2F*). This was observed for all four pulses of input summations delivered at 10 Hz. However, for the SLM input stimulation, the observed paired amplitude was significantly smaller than the linear summation of the two inputs (SLM alone + SuM alone) for the first stimulus (n = 10; T test, p=0.014) (*Figure 2F*). This observation is expected, as the attenuation of distal dendritic SLM inputs causes the peak of the PSP to be delayed relative to the more somatic SuM input. Thus, the SuM input paired with either SR or SLM input stimulation has minor depolarizing effect on the PSP in CA2 PNs. However, the SuM input might have different effect on the SR and SLM inputs depending on the precise timing of their activation.

We also examine the summation ratio for a train of 4 PSPs at 10 Hz from SR and SLM synaptic inputs stimulation with and without simultaneous SuM input stimulation (*Figure 2G–H*). We observed a significant reduction of the summation ratio as measured by the ratio of the n-th pulse to the first (Pn/P1) for both SR (n = 10; repeated-measures ANOVA, p=$2.3\times10^{-4}$) and SLM (n = 10; repeated-measures ANOVA, p=$8.5\times10^{-4}$). This observation that concomitant SuM activity is reducing the level of facilitation of several pulses in a train indicates that the short-term dynamics of the SuM-driven excitation and feedforward inhibition are playing a role to prevent cellular excitation from other inputs.

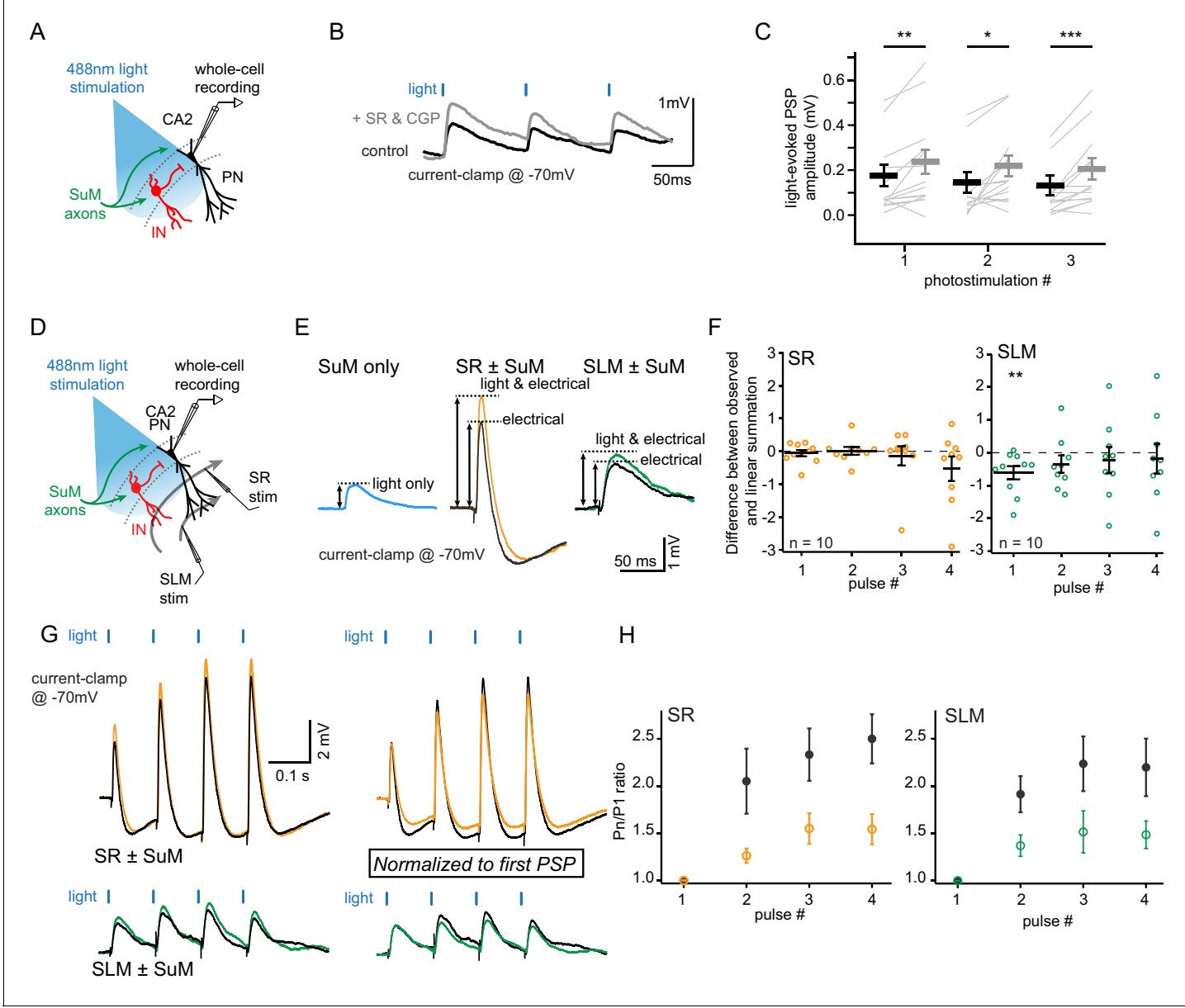

**Figure 2.** SuM input drives inhibition that controls excitation in CA2 PNs. Whole-cell current clamp recordings of light-evoked post-synaptic potentials (PSPs) from SuM input stimulation onto CA2 PNs reveal contribution of feedforward inhibition in dampening excitatory input at −70 mV. (**A**) Diagram illustrating whole-cell recording configuration in acute hippocampal slices. During these experiments, direct current was injected as necessary to maintain a membrane potential of −70 mV. (**B**) Sample traces of three 10 Hz SuM light-evoked PSPs in a CA2 PN before and after blocking inhibitory transmission (control shown in black, SR95531 and CGP55845A in gray). (**C**) Summary graph of light-evoked PSP amplitudes recorded in PNs before and after application of 1 μM SR95531 and 2 μM CGP55845A (individual cells shown as thin lines, population average shown as thick line, error bars represent SEM, n = 14; Wilcoxon signed-rank tests, p=0.004 for the first PSP, p=0.013 for the second PSP, p<0.001 for the third PSP). (**D-H**) Summation of SuM synaptic potentials with SR and SLM electrical input stimulation. (**D**) Diagram illustrating the recording configuration similar to panel A but with stimulating electrodes positioned in *stratum lacunosum moleculare* (SLM) and *stratum radiatum* (SR). (**E**) Left, example traces of PSPs evoked by SuM fiber light stimulation alone (blue trace). Center, PSPs evoked by electrical stimulation of SR inputs alone (black) or paired with simultaneous SuM stimulation (orange). Right, PSPs evoked by electrical stimulation of SLM inputs alone (black) or paired with SuM stimulation (green). (**F**) Plots of the difference between the mathematical summation of the amplitudes of the SuM PSP amplitude and electrical stimulation (linear summation) and the measured SuM +electrical PSP. Left, SR inputs are not significantly different from zero, indicating that SuM and SR inputs linearly summate. Right, for the first pulse, the measured SLM + SuM amplitude is significantly smaller (n = 10; T test, p=0.014) than the expected linear summation. (**G**) Left, example traces of 10 Hz trains of PSPs of either electrical stimulation (black traces) or trains of paired electrical and light stimulation of SuM fibers (SR + SuM in orange or SLM + SuM in green). Right, traces with amplitudes normalized to the first PSP for both the electrical and simultaneous light and electrical PSPs. The amplitudes for all PSPs are measured from the potential immediately before each stimulus. (**H**) Summary plots of the

*Figure 2 continued on next page*

Figure 2 continued

summation ratio of the 2nd, 3rd, and 4th PSP for electrical stimulation (black symbols) or paired stimulation of SR + SuM (left, orange) or SLM + SuM (right, green).

## Basket cells are strongly recruited by the SuM input

Because the hippocampus hosts a variety of interneurons (INs) that are involved in controlling specific aspects of PN excitability, we wished to establish which kind of IN was targeted by the SuM input to area CA2. We performed whole-cell recordings from INs in this area and assessed post-synaptic excitatory responses to SuM axons stimulation in these cells (*Figure 3*). In contrast with previous reports of an exclusive innervation of PNs by SuM (*Maglóczky et al., 1994*), we observed robust light-evoked excitatory transmission from SuM axons in 35 out of 62 interneurons (INs) with soma located in SP. Following biocytin-streptavidin staining and anatomical reconstructions of recorded INs (allowing unequivocal identification in 48 neurons), we were able to classify INs based on their physiological properties, somatic location and axonal arborization location. We classified 22 cells as basket cells (BCs) because their axonal arborizations were restricted to SP (*Figure 3A*). BCs fired APs at high frequency either in bursts or continuously upon depolarizing current injection and showed substantial repolarizing sag current when hyperpolarized (*Figure 4A*, *Table 3*). Light-evoked EPSCs and PSPs were readily observed in the vast majority of BCs (*Figure 3A, C and D*, *Table 4*) and reached large amplitudes in some instances. An additional 26 INs with soma in SP were classified as non-BCs because their axon did not target SP (*Figure 3B*). In our recordings, these cells fired in bursts and showed little sag during hyperpolarizing current injection steps (*Table 3*). We consistently observed no or very minor light-evoked excitatory transmission onto non-BCs (*Figure 3B–C*, *Table 4*). Furthermore, we recorded from 17 INs that had soma in *stratum oriens* (SO) and nine in *stratum radiatum* (SR). Like non-BCs, these INs did not receive strong excitation from SuM fibers (*Table 4*). This data is consistent with the conclusion that SuM input preferentially forms excitatory synapses onto basket cells in area CA2.

To fully assess the strength of SuM inputs onto the different cell types, we examined the following parameters for each population: the connectivity, success rate, amplitude, potency, kinetics, and latencies of EPSCs as well as the resulting depolarization of the membrane potential. First, SuM inputs preferentially innervated BCs as evidenced by a higher connectivity of EPSCs in BCs than in PNs or other INs (*Table 4*). Importantly, excitatory responses had short latencies with limited jitter (*Table 4*) indicating that the connection was monosynaptic in all cell types. When voltage-clamping cells at −70 mV, light-evoked EPSCs could be compared between different cell populations. However, not every photostimulation gave rise to an EPSC leading to an average success rate that tended to be highest in BCs (*Table 4*). In addition, BCs appeared to receive more excitation from the SuM input than other cells types, as the amplitude of EPSCs was larger in BCs than in PNs (*Table 4*). EPSCs recorded in BCs also had faster kinetics than in PNs (*Table 4*). Interestingly, combining the success rate of EPSCs with their respective amplitudes to compute the potency of the SuM synapses revealed that it was significantly larger in BCs than in PNs and non-BCs (*Figure 3C*; potencies were −12 ± 1.6 pA for PNs, n = 166; −29 ± 7.8 pA for BCs, n = 18; −5.9 ± 1.5 pA for non-BCs, n = 13; Kruskal-Wallis test with Dunn-Holland-Wolfe *post hoc* test, p=0.022). Consequently, EPSPs recorded at −70 mV were of larger amplitude in BCs than in PNs and non-BCs (*Figure 3D*; amplitudes were 0.44 ± 0.06 mV for PNs, n = 20; 1.71 ± 0.57 mV for BCs, n = 10; 0.53 ± 0.07 mV for non-BCs, n = 4; Kruskal-Wallis test with Dunn-Holland-Wolfe post hoc test, p<0.001). When recording cell-attached or current-clamping BCs at their resting membrane potential ($V_M$), photostimulation of SuM axons was able to evoke AP firing (*Figure 3E*) in multiple instances (n = 7 of 13). However, this was never observed in PNs (n = 0 of 78), non-BCs (n = 0 of 16), SR INs (n = 0 of 9) or SO INs (n = 0 of 8). These results show that SuM projections to area CA2 preferentially provide excitation to BCs that are likely responsible of the feedforward inhibition observed in PNs. This is in accordance with an efficient control of area CA2 PNs excitation by the SuM inhibitory drive as axons from BCs deliver the feedforward inhibition to the peri-somatic region of PNs, effectively shunting incoming PSPs from both the SuM and from dendritic-targeted inputs in SR and SLM.

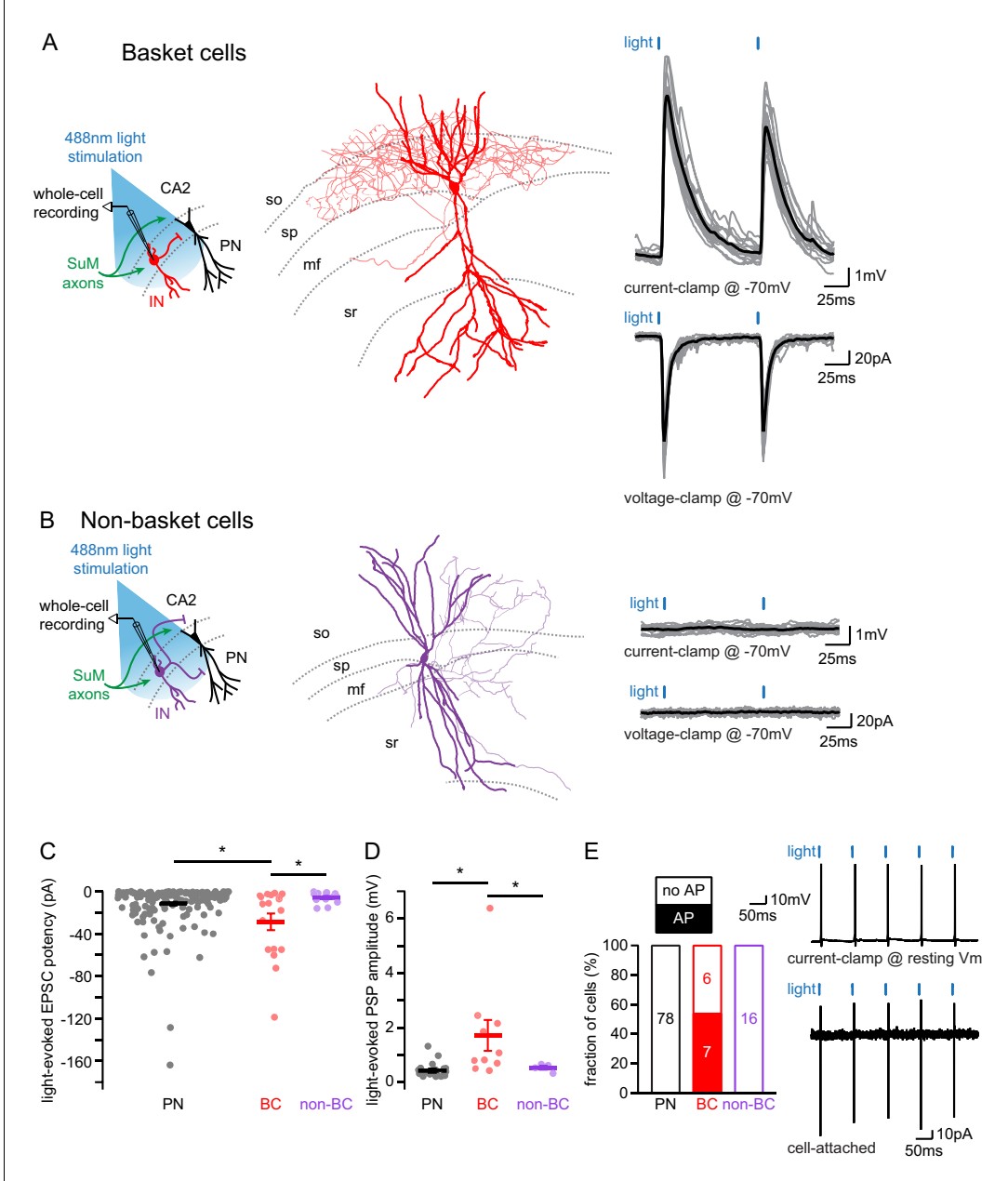

**Figure 3.** SuM input provides strong excitatory glutamatergic transmission to basket cells (BCs) in area CA2. (**A-B**) Left, diagrams illustrating whole-cell recordings in area CA2 and SuM fiber stimulation in acute slice preparation. Middle, example reconstruction of different cell types (soma and dendrites in thick lines, axon in thin lines, hippocampal strata in dotted gray lines). Right, sample traces of light-evoked EPSPs (top, individual traces in gray, average trace in black) and EPSCs (bottom, individual traces in gray, average trace in black). (**A**) Basket cell in area CA2. (**B**) Non-basket cell in area CA2. (**C**) Summary graph of light-evoked EPSC potencies in PNs, BCs, and non-BCs in area CA2 (individual cells shown as dots, population average shown as thick line, error bars represent SEM, PNs: n = 166; BC INs: n = 18; non-BCs: n = 13; Kruskal-Wallis test with Dunn-Holland-Wolfe post hoc test, p=0.022). (**D**) Summary graph of light-evoked PSP amplitudes in PNs, BCs and non-BCs (individual cells shown as dots, population average shown as thick line, error bars represent SEM, PNs: n = 20; BCs: n = 10; non-BCs: n = 4; Kruskal-Wallis test with Dunn-Holland-Wolfe post hoc test, p<0.001). (**E**) Left, proportion of post-synaptic CA2 PNs, BCs and non-BCs firing action potentials time-locked to light stimulation of SuM input. Right, sample traces of light-evoked action potentials in a BC recorded in current-clamp at resting membrane potential (top) and in cell-attached (bottom) configurations.

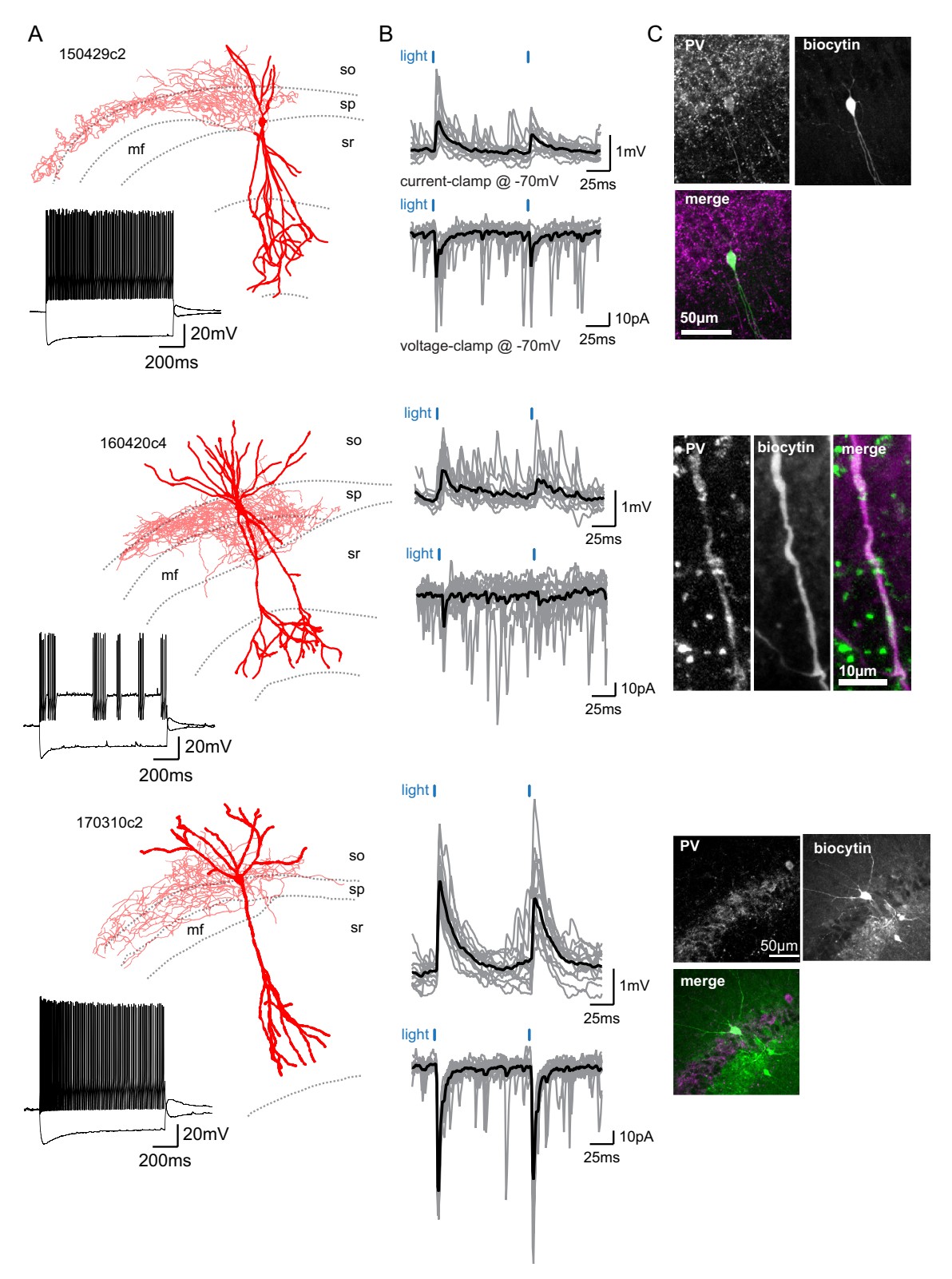

**Figure 4.** SuM input provides excitation to Parvalbumin-expressing BCs. (**A**) Three biocytin reconstructions of BC INs with dendrites in red and axons in light red. Inset, current clamp steps to −400 pA and +400 pA display high-frequency AP firing and repolarizing sag current. (**B**) Corresponding light-evoked EPSCs and EPSPs for the three reconstructed neurons (individual traces in gray, average trace in black). (**C**) Corresponding PV immunostaining of the three interneurons: parvalbumin staining, biocytin labeling of the recorded cell, and merge (PV in magenta and biocytin in green).

**Table 3.** Electrophysiological properties of interneurons in SuM-innervated area.

| | $V_M$ (mV) | $R_M$ (MOhm) | $C_M$ (pF) | Firing adaptation index | Sag (mV) |
|---|---|---|---|---|---|
| Basket cell (n = 16) | −57.3 ± 1.38 | 144 ± 28.1 | 64.0 ± 8.70 | 0.74 ± 0.05 | 9.4 ± 1.0 |
| Non-Basket Cell (n = 12) | −55.6 ± 1.84 | 224 ± 46.8 | 52.0 ± 5.90 | 0.57 ± 0.06 | 5.9 ± 1.4 |
| Interneuron SO (n = 6) | −57.0 ± 3.16 | 201 ± 21.0 | 44.7 ± 5.31 | 0.61 ± 0.11 | 7.6 ± 1.9 |
| Interneuron SR (n = 8) | −60.1 ± 2.89 | 282 ± 49.8 | 39.6 ± 3.18 | 0.65 ± 0.09 | 8.1 ± 2.1 |
| Statistics | One-way ANOVA test p=0.527 | One-way ANOVA test p=0.100 | Kruskal-Wallis test p=0.354 | One-way ANOVA test p=0.238 | One-way ANOVA test p=0.292 |

## Parvalbumin-expressing basket cells mediate the feedforward inhibition driven by SuM

In the hippocampus, BCs express either cholecystokinin (CCK) or parvalbumin (PV) (*Klausberger and Somogyi, 2008*). We found that in response to a 1 s depolarizing pulse, most BCs that received strong SuM excitatory input displayed very fast AP firing with little accommodation in the AP firing frequency (*Figure 4A–B*, *Table 3*). This firing behavior is similar to what has been reported for fast spiking PV-expressing BCs in CA1 (*Pawelzik et al., 2002*). In contrast, CCK-expressing BCs show a lower firing frequency and more accommodation during the train (*Pawelzik et al., 2002*). This result suggests that BCs connected by the SuM may be expressing PV. To directly confirm this hypothesis, we performed post hoc immunostaining of recorded interneurons that received strong excitation from SuM input. Because of the dialysis inherent to the whole-cell recording conditions, we encountered difficulty staining for multiple cells. However, PV-immunoreactivity could unequivocally be detected in either the soma or dendrites of seven connected BCs (*Figure 4C*). Therefore, this data demonstrates that at least a fraction of the recorded BCs connected by the SuM is expressing PV.

Hence, to address whether the lack of PV staining in some cells was a consequence of dialysis or resulted from the fact that non-PV +BCs are also connected, we made use of different strategies to differentiate PV+ and CCK+ INs. First, we wished to genetically confirm that PV+ INs are involved in the SuM-driven feedforward inhibition of area CA2 PNs. We used inhibitory Gi-DREADD to selectively inhibit PV+ INs in area CA2 while monitoring feedforward IPSCs from area CA2 PNs in response to SuM stimulation. To achieve that, we injected AAVs expressing a Cre-dependent hM4D (Gi) inhibitory Gi-DREADD in area CA2 of PV-Cre mice together with AAVs expressing ChR2 with a pan-neuronal promoter in the SuM (*Figure 5A*). While we were able to obtain very specific expression of DREADD in PV+ INs, only a fraction of PV+ INs had detectable DREADD expression as quantified by immunohistochemistry (*Figure 5B*; fraction of PV+ INs expressing DREADDs in CA2 = 75 ± 3.5%, n = 13). We observed a substantial reduction of SuM-evoked IPSC amplitude recorded in area CA2 PNs upon application of 10 µM of the Gi-DREADD ligand CNO (*Figure 5C*; amplitudes were 847 ± 122 pA in control and 498 ± 87 pA in CNO hence a 42 ± 6.0% block, n = 13; paired-T test, p<0.001). Although we never measured a complete block of inhibitory responses, this result

**Table 4.** Characteristics of excitatory SuM light-evoked transmission onto interneurons and pyramidal cells.

| Cell type | Connectivity (%) | Amplitude (pA) | Rise time (ms) | Decay time (ms) | Latency (ms) | Success rate |
|---|---|---|---|---|---|---|
| Pyramidal cell | 63 (n = 166 of 263) | −19 ± 1.6* | 3.4 ± 0.1* | 15 ± 0.5* | 2.9 ± 0.1 | 0.46 ± 0.02 |
| Basket cell | 82 (n = 18 of 22) | −43 ± 8.7* | 1.7 ± 0.3* | 8.4 ± 1.3* | 3.1 ± 0.4 | 0.59 ± 0.07 |
| Non-Basket Cell Interneuron SO Interneuron SR | 39 (n = 10 of 26) 12 (n = 2 of 17) 11 (n = 1 of 9) | −16 ± 2.8 | 2.6 ± 0.5 | 12 ± 1.4 | 3.4 ± 0.7 | 0.36 ± 0.06 |
| Statistics | $\chi^2$ test p=0.006* | Kruskal-Wallis test p=0.016 Dunn-Holland-Wolfe post hoc p<0.05* | One-way ANOVA test p<0.001 Tukey post hoc p<0.001* | One-way ANOVA test p<0.001 Tukey post hoc p<0.001* | One-way ANOVA test p=0.580 | One-way ANOVA test p=0.066 |

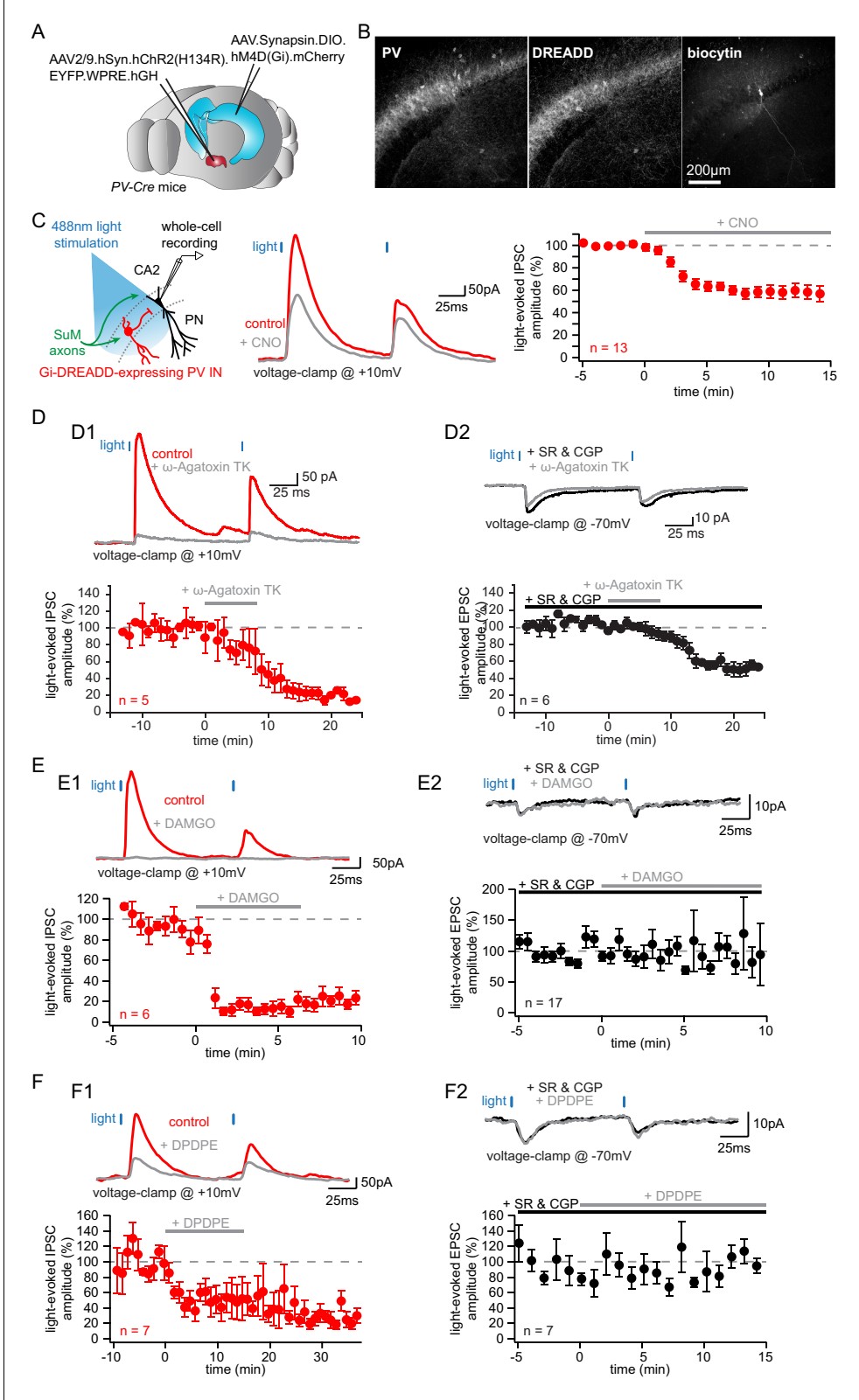

**Figure 5.** Parvalbumin-expressing BCs mediate the feedforward inhibition recruited by photostimulation of SuM fibers. (**A-C**) Silencing of PV+ INs by inhibitory DREADDs reduces SuM feedforward inhibition onto area CA2 PNs. (**A**) Diagram illustrating the method to infect SuM neurons and selectively inhibit PV+ INs in area CA2. An AAV allowing the Cre-dependent expression of inhibitory DREADD was injected bilaterally into area CA2 of the dorsal hippocampus and another AAV allowing the expression of ChR2 was injected into the SuM of PV-Cre mice, allowing optogenetic stimulation of SuM

*Figure 5 continued on next page*

*Figure 5 continued*

inputs and pharmacogenetic inhibition of PV+ INs by application of the DREADD agonist CNO at 10 µM. (**B**) Example immunostaining against PV, DREADD and biocytin labeling in area CA2 from a slice used in these experiments. (**C**) Left, diagram of the recording configuration in hippocampal slices. Center, sample traces (control in red, CNO in gray). Right, summary graph of light-evoked IPSC amplitudes recorded in CA2 PNs before and after application of 10 µM CNO (n = 13, error bars represent SEM). (**D**) Application of the P/Q-Type voltage-activated calcium channel blocker ω-agatoxin TK results in nearly complete loss of feedforward inhibition recruited by light activation of SuM inputs in area CA2. D1, sample traces, (top, control in red, ω-agatoxin TK in gray) and summary graph of light-evoked IPSC amplitudes recorded in CA2 PNs before and after application of 200 nM ω-agatoxin TK (bottom, n = 5, error bars represent SEM). D2, sample traces (top, SR95531 and CGP55845A in black, ω-agatoxin TK in gray) and summary graph of light-evoked EPSC amplitudes before and after application of 200 nM ω-agatoxin TK (bottom, n = 6, error bars represent SEM). (**E**) Application of the mu-opioid receptor agonist, DAMGO, results in the complete abolition of light-evoked SuM inhibitory transmission. E1, sample traces (top, control in red, DAMGO in gray) and summary graph of light-evoked IPSC amplitudes recorded in CA2 PNs before and after application of 1 µM DAMGO (bottom, n = 6, error bars represent SEM). E2, sample traces (top, SR95531 and CGP55845A in black, DAMGO in gray) and summary graph of light-evoked EPSC amplitudes before and after application of 1 µM DAMGO (bottom, n = 17, error bars represent SEM). (**F**) Application of the delta-opioid receptor agonist, DPDPE, results in the long-term depression of light-evoked SuM inhibitory transmission. F1, sample traces (top, control in red, DPDPE in gray) and summary graph of light-evoked IPSC amplitudes before and after application of 0.5 µM DPDPE (bottom, n = 7, error bars represent SEM). F2, sample traces (top, SR95531 and CGP55845A in black, DAMGO in gray) and summary graph of light-evoked EPSC amplitudes before and after application of 0.5 µM DPDPE (bottom, n = 7, error bars represent SEM).

The online version of this article includes the following figure supplement(s) for figure 5:

**Figure supplement 1.** Evidence that Gi-DREADD expression in PV+ cells is not sufficient to prevent action potential firing.

unequivocally places PV+ INs as mediators of the SuM feedforward inhibition of area CA2 PNs. The incomplete block of IPSCs observed in these experiments indicates that either additional non-PV + INs are recruited by SuM input or that our silencing of PV-mediated feedforward inhibition is incomplete. This could be a consequence of partial infection of PV +INs in area CA2 by AAVs carrying DREADDs and partial silencing of DREADD-expressing PV+ INs by CNO. To address the latter, we performed whole-cell recordings from Gi-DREADD-expressing CA2 PV+ INs labeled with mCherry and monitored the variations in $V_M$ level and action potential firing to SuM input stimulation before and after CNO application (*Figure 5—figure supplement 1A*). We found that CNO application caused a significant hyperpolarization of Gi-DREADD-expressing CA2 PV+ INs, albeit modest in magnitude (*Figure 5—figure supplement 1B–D*; $V_M$ were −55.3 ± 2.3 mV in ACSF and −61.8 ± 2.7 mV in CNO hence a −6.5 ± 2.4 mV hyperpolarization, n = 6; Wilcoxon signed-rank test, p=0.031). While this confirmed the relevance of our silencing strategy, it highlighted the possibility that Gi-DREADD-expressing CA2 PV+ INs may not be fully silenced by CNO. Indeed, we observed residual SuM-evoked AP firing in these cells after CNO application (*Figure 5—figure supplement 1D–E*). These data indicate that synaptically evoked somatic AP firing is not fully blocked by CNO in Gi-DREADD-expressing CA2 PV+ INs. Because it is difficult to distinguish between partial silencing of PV INs by Gi-DREADDs or recruitment of other types of INs in the SuM-driven feedforwards inhibition, we adopted other complementary strategies to answer this question.

We used a pharmacological strategy to selectively manipulate PV+ INs by targeting their GABA release machinery. In the neocortex, P/Q-type voltage-gated calcium channels are necessary for GABA release from PV+ fast spiking INs onto PNs (*Zaitsev et al., 2007*). In contrast, N-type calcium channels are primarily involved in GABA release from CCK+ INs (*Hefft and Jonas, 2005*). Thus, we recorded SuM input-evoked EPSCs and IPSCs in CA2 PNs before and after application of the P/Q-type voltage-gated calcium channels specific blocker ω-agatoxin TK (200 nM) (*Figure 5D*). We observed a near-complete block of IPSCs upon ω-agatoxin TK application (*Figure 5D* **1**, IPSC amplitudes were 245.5 ± 92.6 pA in control and 35.0 ± 15.4 pA in ω-agatoxin TK hence a 81.8 ± 3.9% block, n = 5; paired-T test, p<0.001), suggesting a major contribution from PV+ INs to SuM-driven feedforward inhibition consistent with our previous results. However, we observed that excitatory transmission from SuM axons was also partially blocked by ω-agatoxin TK application, as SuM input-evoked EPSCs were significantly reduced although not abolished (*Figure 5D* **2**, EPSC amplitudes were −51.8 ± 5.9 pA in SR95531 and CGP55845A and −26.5 ± 5.4 pA after ω-agatoxin TK hence a 49.6 ± 5.6% block, n = 6; paired-T test, p<0.001). This observation indicates that glutamate release from SuM axons relies on P/Q-type voltage-gated calcium channels to some degree, thereby complicating the interpretation of the reduction of IPSC amplitude in CA2 PNs.

It has previously been demonstrated that PV+ BC transmission can be strongly attenuated by mu opioid receptor activation (MOR) while CCK+ BC transmission is insensitive to MOR activation (*Glickfeld et al., 2008*). Thus, we recorded from PNs in area CA2 and examined the sensitivity of light-evoked IPSCs to the application of the MOR agonist DAMGO (*Figure 5E*). We found that there was a near complete block of the light-evoked IPSC amplitude following 1 µM DAMGO application (*Figure 5E* **1**; IPSC amplitudes were 343 ± 123 pA in control and 31 ± 12.4 pA in DAMGO hence a 88 ± 5.0% block, n = 6 PNs; Wilcoxon signed-rank test, p=0.031), while direct excitatory transmission remained unaffected (*Figure 5E* **2**; EPSC amplitudes were −6.7 ± 1.1 pA in SR95531 and CGP55845A and −5.6 ± 0.9 pA after DAMGO, n = 17 PNs; Wilcoxon signed-rank test, p=0.19). While this result is in agreement with our DREADD and ω-agatoxin TK results showing a major contribution of PV +INs to the SuM-driven feedforward inhibition, it should be noted that the dichotomy between PV+ versus CCK+ INs sensitivity to DAMGO has not been directly verified in area CA2.

It has recently been shown that delta opioid receptors (DORs) are specifically expressed in a fraction of PV+ cells in the hippocampus (*Erbs et al., 2012*). Furthermore, PV+ INs in area CA2 are the substrate of an iLTD of feedforward inhibition from CA3 mediated by delta opioid receptor (DOR) activation (*Nasrallah et al., 2019*; *Piskorowski and Chevaleyre, 2013*). Therefore, we sought to further refine our characterization of the SuM feedforward inhibition by assessing its sensitivity to DOR activation (*Figure 5F*). Application of 0.5 µM of the DOR agonist DPDPE led to a long-term reduction of light-evoked IPSCs recorded in area CA2 PNs, similar to the iLTD seen by CA3 input stimulation (*Figure 5F* **1**; IPSC amplitudes were 168 ± 28 pA in control and 64 ± 22 pA in DPDPE hence a 61 ± 14% block by DPDPE, n = 7; paired-T test, p=0.015), while leaving direct EPSCs unaffected (*Figure 5F* **2**; EPSC amplitudes were −4.0 ± 1.6 pA in SR95531 and CGP55845A and −3.1 ± 1.1 pA after DPDPE, n = 7; Wilcoxon signed-rank test, p=0.22). Further confirming the PV+ nature of INs responsible for the SuM feedforward inhibition, this result reveals that both the local CA3 and long-range SuM inputs converge onto an overlapping population of INs to inhibit area CA2 PNs, thus enabling cross-talk between these routes through synaptic plasticity of PV+ INs. However, since DORs are only expressed in a fraction of PV+ INs and therefore only reduces but does not fully block PV+ IN-mediated GABA release (*Nasrallah et al., 2019*; *Piskorowski and Chevaleyre, 2013*), it is difficult to know if the remaining SuM-evoked IPSCs are from PV+ INs not expressing DOR or from other INs recruited by the SuM input.

Altogether, these four methods strongly suggest that SuM inputs selectively recruit PV +interneurons to inhibit CA2 PNs. Although individually each method does not conclusively demonstrate that SuM input exclusively targets PV+ INs, the consistent reduction of SuM-driven feedforward inhibition of CA2 PNs observed with every approach allows us to conclude that PV + cells are predominantly targeted by SuM inputs in area CA2.

## The feedforward inhibitory drive from SuM controls pyramidal neuron excitability

Given SuM axonal stimulation triggers an excitatory-inhibitory sequence in post-synaptic PNs, we asked which effect would prevail on PN excitability. In order to assess this, we mimicked an active state in PNs by injecting constant depolarizing current steps sufficient to sustain AP firing during 1 s while photostimulating SuM axons at 10 Hz (*Figure 6A–B*). We observed that recruitment of SuM inputs significantly delayed the onset of the first AP (*Figure 6C*; latency to the first AP were 221 ± 19.9 ms in control and 233 ± 19.1 ms with photostimulation, hence a 12.1 ± 4.3 ms increase upon photostimulation, n = 12; paired-T test, p=0.016). In addition, given SuM neurons display theta-locked firing in vivo, we asked if rhythmic inhibition driven by SuM inputs in area CA2 could pace AP firing in PNs by defining windows of excitability. Indeed, photostimulation of SuM axons at 10 Hz led to a significant decrease of variability in the timing of AP firing by PNs (*Figure 6D–E*; standard deviations of the first AP timing were 36.9 ± 11 ms in control and 24.7 ± 7.4 ms with photostimulation, hence a 12.3 ± 5.3 ms decrease upon photostimulation, n = 12; Wilcoxon signed-rank tests, p<0.001 for the first AP, p=0.008 for the second AP, p=0.004 for the third AP). Both the delay of AP onset and the reduction of AP jitter stemmed from the feedforward inhibition recruited by SuM inputs as application of GABA_A and GABA_B receptor antagonists abolished these effects of SuM stimulation (*Figure 6C–E*; latency to the first AP were 232 ± 19.8 ms in SR95531 and CGP55845A and 235 ± 18.0 ms with photostimulation, n = 6; Wilcoxon signed-rank test, p=0.44; standard deviations of the first AP timing were 11.9 ± 2.0 ms in SR95531 and CGP55845A and

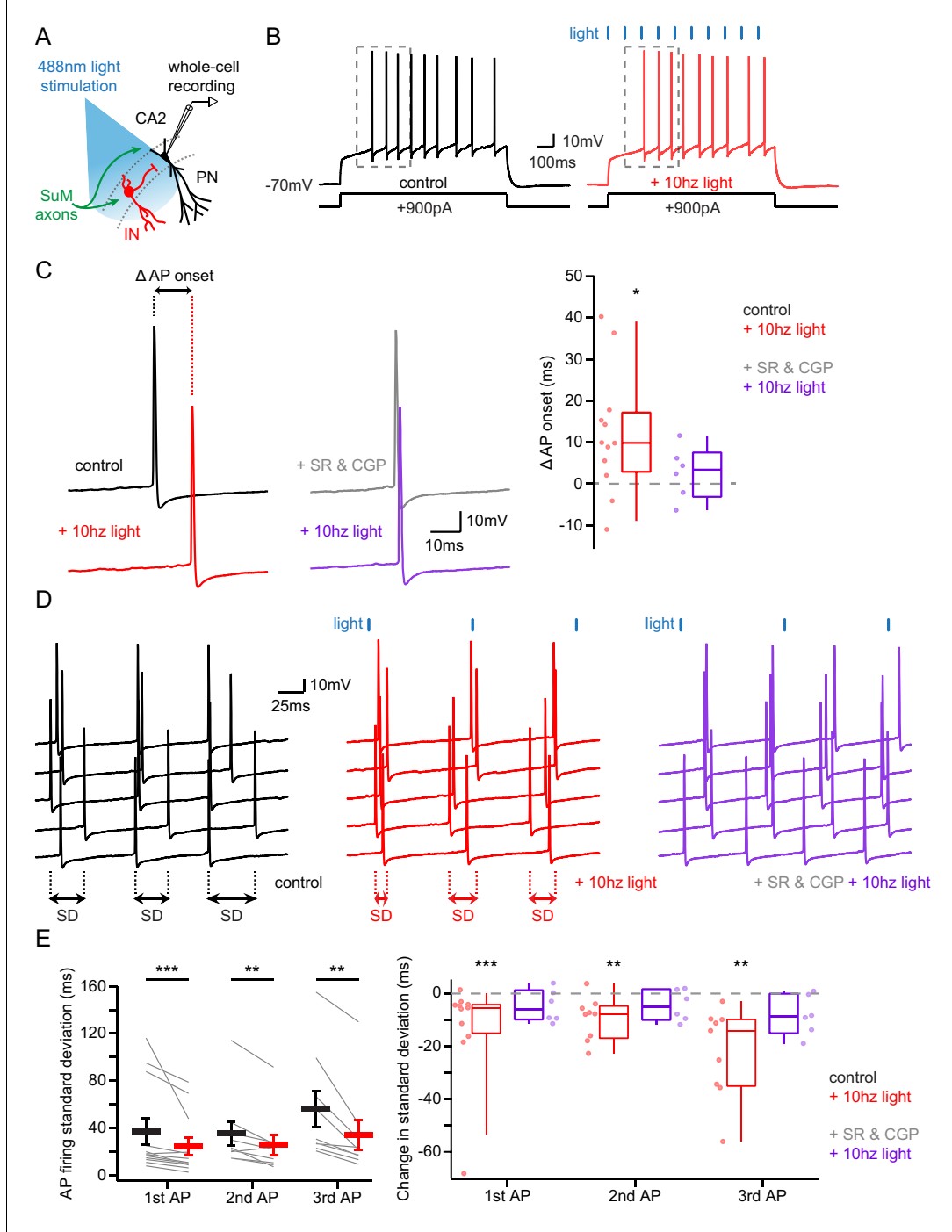

**Figure 6.** Area CA2 PNs receive a net inhibitory drive from SuM that controls AP firing properties. (**A**) Diagram illustrating whole-cell recordings of area CA2 PNs and SuM fiber light stimulation in acute slice preparation. (**B**) Example traces of a CA2 PN action potential firing in response to current injection in the absence (black traces) or presence of 10 Hz photostimulation of SuM inputs (red traces). (**C**) Action potential onset latency is increased with 10 Hz SuM input photostimulation. Left, sample traces of the first AP in control and with inhibition blocked by 1 μM SR95531 and 2 μM CGP55845A application (light-off in black, light-on in red, light-off in SR95531 and CGP55845A in gray, light-on in SR95531 and CGP55845A in purple). Right, summary graph of photostimulation-induced delay of AP firing in area CA2 PNs before and after application of SR95531 and CGP55845A (control shown in red, n = 12, paired-T test, p=0.016; SR95531 and CGP55845A shown in purple, n = 6; Wilcoxon signed-rank test, p=0.44; individual cells shown with dots, boxplot represents median, quartiles, 10th and 90th percentiles). (**D**) Sample traces of AP firing in repeated trials (light-off in black, light-on in red, light-on in SR95531 and CGP55845A in purple; during experiment photostimulation was interleaved with control but traces are grouped here for demonstration purposes). (**E**) AP jitter in CA2 PNs is reduced by activation of SuM inputs. Left, summary graph of the standard deviation of AP firing with or without 10 Hz photostimulation (n = 12; Wilcoxon signed-rank test, p<0.001 for the first AP, p=0.008 for the second AP, p=0.004 for the

*Figure 6 continued on next page*

**Figure 6 continued**

third AP; individual cells shown with thin lines, population average shown as thick line, error bars represent SEM). Right, photostimulation-induced reduction of AP firing standard deviation in control and in SR95531 and CGP55845A (control, n = 12; Wilcoxon signed-rank tests, p<0.001 for the first AP, p=0.008 for the second AP, p=0.004 for the third AP; SR95531 and CGP55845A, n = 6; Wilcoxon signed-rank tests, p=0.22 for the first AP, p=0.16 for the second AP, p=0.09 for the third AP; individual cells shown with dots, boxplot represents median, quartiles, 10th and 90th percentiles).

7.1 ± 1.5 ms with photostimulation, n = 6; Wilcoxon signed-rank tests, p=0.22 for the first AP, p=0.16 for the second AP, p=0.09 for the third AP). These results reveal that the purely glutamatergic SuM input, by recruiting feedforward inhibition, has an overall inhibitory effect on PN excitability and can influence the timing and jitter of area CA2 PN action potential firing.

One drawback of these results is that the injection of current steps to evoke action potential firing is not entirely representative of CA2 PN activity, as there is no synaptic input leading to AP firing. It has been reported that the AP discharge of SuM neurons in vivo is phase-locked to the hippocampal theta rhythm (*Kocsis and Vertes, 1994*). Because theta rhythm is a brain state characterized by elevated levels of acetylcholine, we approximately mimicked these conditions in the hippocampal slice preparation by bath application of 10 μM of the cholinergic agonist carbachol (CCh). Under these conditions, CA2 PNs depolarize and spontaneously fire rhythmic bursts of APs, and the properties of these AP bursts are tightly controlled by excitatory and inhibitory synaptic transmission (*Robert et al., 2020*). Thus, we decided to examine how SuM input stimulation influenced CA2 PN firing under these conditions.

Because SuM neurons fire in bursts at theta frequency in vivo (*Kirk et al., 1996*), and because the elevated cholinergic tone accompanying theta can activate muscarinic receptors that alter the synaptic release properties of many synapses in the brain, we examined how synaptic transmission from the SuM input to area CA2 was affected by the application of 10 μM carbachol (CCh) (*Figure 7—figure supplement 1A*; *Kirk et al., 1996*; *Kocsis and Vertes, 1994*). With GABA receptors blocked to first assess the SuM excitatory transmission only, we observed that CCh decreased the amplitude and increased the PPR of SuM-evoked EPSCs in CA2 PNs (*Figure 7—figure supplement 1B*). This suggests a decrease of glutamate release by SuM axons induced by CCh. We found similar results for SuM-evoked feedforward inhibitory transmission to CA2 PNs as IPSC amplitude was decreased and PPR increased with CCh application (*Figure 7—figure supplement 1C*). Next, we examined the relative short-term dynamics of SuM-evoked excitatory and inhibitory transmission to CA2 PNs. For this, both EPSCs and IPSCs were recorded from the same individual CA2 PNs upon repeated SuM input stimulation with five pulses delivered at 10 Hz before and after CCh application (*Figure 7—figure supplement 1D–G*). We observed that both SuM-evoked EPSCs and IPSCs underwent short-term depression, as evidenced by a decrease in amplitude along the pulse train as well as amplitude ratios between subsequent pulses over the first pulse (Pn/P1) (*Figure 7—figure supplement 1D–F*, *Figure 7—figure supplement 1—source data 1*). It is worth noting that the Pn/P1 ratio was similar for EPSCs and IPSCs and that the E/I ratio did not significantly change with repeated SuM input stimulation (*Figure 7—figure supplement 1F–G*, *Figure 7—figure supplement 1—source data 1*). This indicates that the SuM influence over CA2 PN may remain overall inhibitory during prolonged SuM input activation. Similarly influencing both EPSCs and IPSCs, application of 10 μM CCh affected these short-term dynamics of the SuM-CA2 PN transmission by decreasing the amplitude of the initial response (*Figure 7—figure supplement 1D–E*, *Figure 7—figure supplement 1—source data 1*) but limiting the subsequent short-term depression of SuM-evoked PSCs amplitude (*Figure 7—figure supplement 1D–F*, *Figure 7—figure supplement 1—source data 1*). Interestingly, the overall effect of repeated SuM input stimulation on post-synaptic responses in area CA2 PNs was even more biased toward inhibition after CCh application as the E/I ratio of PSCs during the pulse train was lower in CCh as compared to control (*Figure 7—figure supplement 1G*, *Figure 7—figure supplement 1—source data 1*), possibly because of a lesser depression of IPSCs as compared to EPSCs (*Figure 7—figure supplement 1D–F*, *Figure 7—figure supplement 1—source data 1*) which could be due to a CCh-induced depolarization of INs mediating SuM-evoked feedforward inhibition. Altogether, these observations match with our findings of the SuM input having an overall inhibitory influence over area CA2, and suggest that this effect might be more gradual over time but even stronger in conditions of elevated cholinergic tone.

Under these conditions of elevated cholinergic tone, we asked how the spontaneous AP bursting activity of CA2 PNs would be affected by activation of the SuM input by triggering 10 s-long trains of 0.5 ms light pulses delivered at 10 Hz to stimulate SuM axons at the onset of bursts (*Figure 7A*). Because of the intrinsic cell-to-cell variability of bursting kinetics, we photostimulated SuM inputs only during interleaved bursts in the same cells. To do this, bursts were detected automatically with

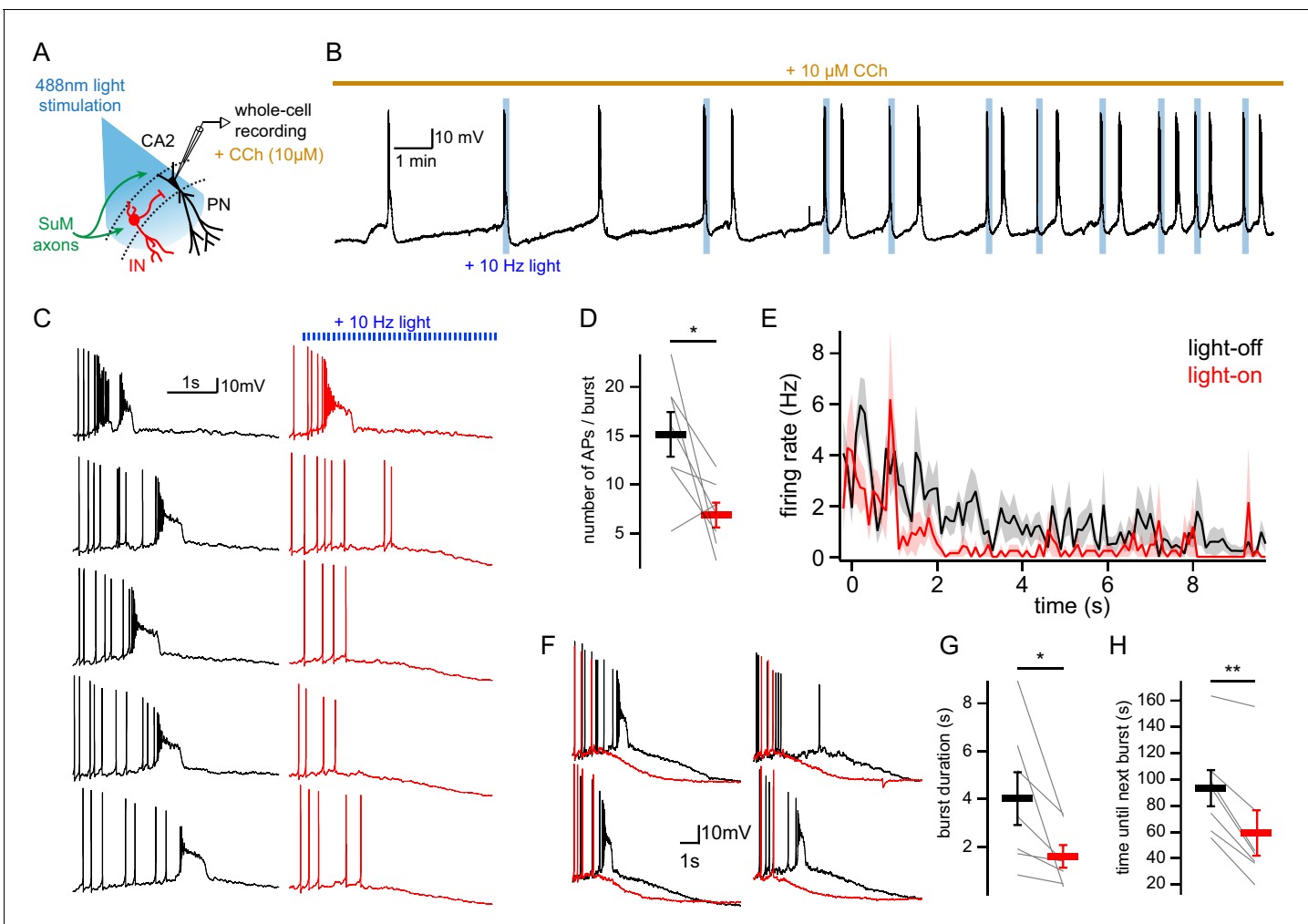

**Figure 7.** SuM input shapes CA2 PN AP bursts in conditions of elevated cholinergic tone. (A) Diagram illustrating whole-cell recordings of area CA2 PNs with light stimulation of SuM fibers in an acute slice preparation. (B) Sample trace of spontaneous AP bursting activity recorded from a CA2 PN during bath application of 10 µM CCh. For every even-numbered burst, a 10 Hz photostimulation (blue bars) was delivered to excite SuM inputs in area CA2 allowing a comparison of burst AP firing in the same cell. (C) Sample traces of AP firing during bursts for light-off (left, black) and light-on (right, red) epochs. (D) Comparison of AP number per burst for light-off (black) and light-on (red) events (n = 7; individual cells shown as thin lines, population average shown as thick line, error bars represent SEM; paired-T test, p=0.031). (E) Average firing rate during spontaneous burst events with SuM photostimulation (red, light-on) and controlled interleaved burst events (black, light-off). Shaded area represents SEM for seven cells each with between 3 and 13 bursts analyzed in light-on and light-off conditions (two-way ANOVA, light factor: p<0.001, time factor: p<0.001, light x time factor: p=0.052). (F) Example burst events with (red) and without (black) SuM photostimulation overlayed and on a scale that shows the rapidly hyperpolarizing membrane potential that occurs with SuM input stimulation. (G) Comparison of bursts duration for events with (red) and without (black) photostimulation (n = 7; individual cells shown as thin lines, population average shown as thick line, error bars represent SEM; paired-T test, p=0.037). (H) Comparison of time elapsed to next burst onset following bursts with (red) or without (black) photostimulation (n = 7; individual cells shown as thin lines, population average shown as thick line, error bars represent SEM; paired-T test, p=0.001).

The online version of this article includes the following source data and figure supplement(s) for figure 7:

**Figure supplement 1.** Effect of carbachol on SuM-CA2 transmission.

**Figure supplement 1—source data 1.** Values for amplitude, Pn/P1 ration and E/I ratio with statistical comparisons for values measured and plotted in *Figure 7—figure supplement 1*.

an online threshold detection system that started the photostimulation pulse train after the first AP of every alternating burst, starting with the second burst (*Figure 7A–B*). For analysis, the number of APs and bursting kinetics could be compared within the same cell. We observed a significant decrease in the number of APs fired during a burst when SuM inputs were photostimulated as compared to interleaved control bursts (*Figure 7C–D*; numbers of APs per burst were 15.2 ± 2.3 in control and 6.9 ± 1.3 with photostimulation, n = 7; paired-T test, p=0.031). In control bursts, the AP firing rate of CA2 PNs initially increases, and then progressively decreases. In the photostimulation bursts, the initial increase of AP firing frequency was absent, and the subsequent AP firing frequency was reduced (*Figure 7E*; two-way ANOVA on firing rate over time in light-on vs light-off conditions; light factor, p<0.001; time factor, p<0.001; light x time factor, p=0.052).

In the presence of CCh, CA2 PNs undergo a depolarization of the membrane potential that is followed by a period of AP firing as the membrane potential remains depolarized for several seconds, and then slowly hyperpolarizes until the next bursting event (*Robert et al., 2020*). We observed that photostimulation of SuM inputs resulted in a striking reduction in the amount of time the membrane potential remained depolarized, and this is likely why the burst duration was significantly shorter in bursts with SuM photo-stimulation (*Figure 7F–G*; burst duration was 4.0 ± 1.1 s in control and 1.6 ± 0.5 s with photostimulation, n = 7; paired-T test, p=0.037). The rate and level of $V_M$ repolarization following bursts were not significantly changed by SuM input photostimulation ($V_M$ repolarization rate was −3.3 ± 0.6 mV/s in control and −3.6 ± 0.7 mV/s with photostimulation, n = 7; paired-T test, p=0.601; post-burst $V_M$ was −62.8 ± 1.7 mV in control and −62.0 ± 2.0 mV with photostimulation, n = 7; paired-T test, p=0.173); however, the inter-burst time interval was reduced. Indeed, AP bursts with SuM input activation were followed more rapidly by another burst of APs than the ones without SuM input activation (*Figure 7B and H*; time until next burst was 93 ± 14 s in control and 59 ± 17 s with photostimulation, n = 7; paired-T test, p=0.001), which could be due to both short-term depression of inhibitory transmission after repeated activation during the SuM input photostimulation train and reduced activation of hyperpolarizing and shunting conductances during bursts shortened by SuM input photostimulation. Thus, in our preparation, SuM input activation is able to modify the spontaneous bursting activity of CA2 PNs under conditions of high cholinergic tone.

As SuM input controls burst firing of action potentials and likely paces activity in area CA2, we wondered how the subsequent output of CA2 PNs would affect their post-synaptic targets. Because CA2 PNs strongly project to CA1 PNs, this activity is likely to influence CA1 encoding and hippocampal output. Thus, we examined the consequences of SuM-CA2 input stimulation on area CA1 both in vivo and in acute slices treated with CCh to induce spontaneous activity (*Figure 8*).

ChR2-EYFP was expressed in the SuM of Csf2rb2-Cre mice in a Cre-dependent manner and the mice were implanted with a microdrive targeting tetrodes to region CA1 and an optical fiber to the SuM terminals in CA2 (*Figure 8A*). Mice were placed in a small box (familiar context) and left free to explore as blue (473 nm) laser light pulses (50 ms pulse width) were applied to the SuM terminals at 10 Hz. Across 23 recording sessions in five mice, we found that the activation of SuM terminals in CA2 resulted in a significant and reproducible change in the multiunit spiking activity recorded in the pyramidal cell layer of CA1 on 34 of 55 tetrodes. The firing rate change was similar across individual tetrodes (*Figure 8B–C*), with a decrease in the normalized firing rate starting shortly after laser onset and continuing for about 10 ms, followed immediately by a rebound-like increase to about 20% greater than baseline firing rate (*Figure 8B–C*).

In order to get a better mechanistic understanding of this observation, we set out to decipher how SuM activity in area CA2 influences CA1 in the hippocampal slice preparation. To this end, we used the same photostimulation protocol used in vivo that consisted of light stimulation trains of 50 ms-long pulses delivered at 10 Hz for 1 s, repeated every 10 s for 2 min and interleaved with light-off sweeps of the same duration, with the microscope objective centered on area CA2. Whole-cell patch-clamp recordings of CA1 PNs were obtained in acute hippocampal slices superfused with CCh and subjected to this light stimulation protocol (*Figure 8D*). We asked what synaptic events may be responsible for the decreased firing of CA1 units observed 10–20 ms after light onset in vivo (*Figure 8A–C*). Whole-cell recordings of CA1 PNs showed an absence of EPSCs time-locked to the photostimulation in all but one case (n = 11/12) (*Figure 8E–F*). In contrast, we often (n = 7/12) observed light-evoked IPSCs in CA1 PNs occurring 10–20 ms after light onset (*Figure 8G–H*). Therefore, the reduction in firing of CA1 units in vivo is likely caused by increased inhibitory inputs onto CA1 PNs within 10–20 ms of SuM fiber stimulation over area CA2. This result highlights a

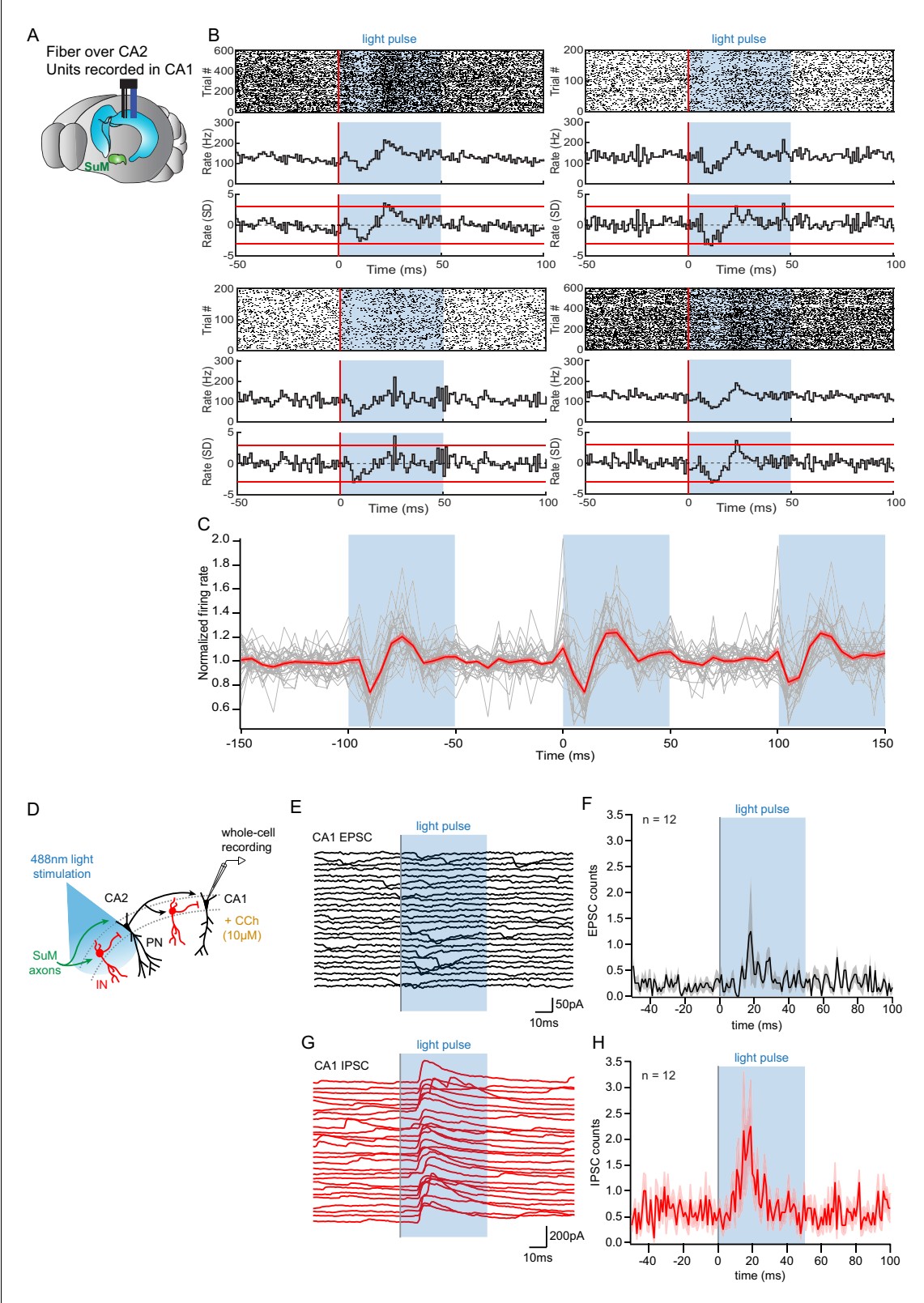

**Figure 8.** Consequences of SuM input on area CA2 output to CA1. (**A**) Diagram illustrating in vivo recording in CA1 with tetrodes and SuM axon terminals stimulation over CA2 with an implanted optical fiber. (**B**) Representative data from four multi-unit recordings. Raster plot (top) showing CA1 AP firing activity before and during photostimulation of SuM fibers in area CA2. The corresponding firing rate histogram (middle) of four tetrodes placed in the CA1 pyramidal cell layers, as well as plots of standard deviation (SD; bottom). Red lines indicate ±3 SD. (**C**) Individual (gray) and average

*Figure 8 continued on next page*

*Figure 8 continued*

(red) normalized firing rates from 34 multiunit recordings, three consecutive light stimulation epochs are displayed to help visualizing the consistency of the effect of SuM input light stimulation over area CA2 on CA1 multi-unit firing; the shaded area represents the SEM. (**D**) Diagram illustrating whole-cell recordings of area CA1 PNs and SuM fiber light stimulation over area CA2 in acute slice preparation. (**E-H**) Example waterfall plots (**E, G**) and corresponding peri-stimulus time histogram (**F, H**) (population average shown as thick line, shaded area represents SEM) of EPSCs (black) and IPSCs (red) recorded from a CA1 PN ex vivo during photostimulation of SuM input over area CA2 with bath application of 10 µM CCh.

contribution of SuM input to controlling CA2 output that regulate CA1 activity in vivo and provides a mechanistic interpretation of this observation at the circuit level.

## Discussion

In this study, we provide direct evidence for a functional connection between the hypothalamus and the hippocampus. Using stereotaxic injection of viral vectors in combination with transgenic mouse lines to express channelrhodopsin in a projection-specific manner, we have been able to selectively stimulate SuM axons in area CA2 of the hippocampus, allowing for the direct examination of synaptic transmission. This approach yielded novel functional information about the SuM input post-synaptic targets and the overall physiological consequences of its activation. We found that, in contrast to previous anatomical reports, SuM input forms synapses onto both PNs and INs in area CA2. The excitatory drive evoked by light-stimulation of SuM input was significantly larger for BC INs, which we demonstrate are likely PV+. The resulting feedforward inhibition recruited by SuM input stimulation enhanced the precision of AP timing of CA2 PNs in conditions of low and high cholinergic tone relevant to different brain states. The modified CA2 output evoked poly-synaptic inhibition in area CA1, likely responsible for a decrease in firing rate of CA1 units in vivo. Overall, we demonstrate that SuM input controls CA2 output to area CA1 by recruiting feedforward inhibition.

### SuM input to area CA2 forms a microcircuit where PV+ basket cells strongly inhibit pyramidal neurons

Glutamatergic innervation of area CA2 by the SuM has been previously described by tracing studies (*Kiss et al., 2000*; *Soussi et al., 2010*) and presumed to form synapses exclusively onto PNs (*Maglóczky et al., 1994*). Our experimental strategy allowed for the direct examination of the post-synaptic targets of SuM glutamatergic axons. Our results confirm that PNs in area CA2 indeed receive excitatory synapses from SuM axons. However, in contrast to what had been proposed in previous studies, we observed that SuM inputs target not only PNs but also INs in area CA2. Importantly, we identified a specific subpopulation of INs as BCs which were the cell type most potently excited by SuM. These BCs could fire action potentials upon SuM input photostimulation leading to a substantial feedforward inhibition of neighboring PNs. We found that at resting membrane potentials, the mixed excitatory/inhibitory SuM input resulted in a net depolarization of the membrane potential in CA2 PNs. However, when the SuM input was paired with either inputs in SR or SLM, we observed a decrease in the summation ratios of trains of synaptic inputs consistent with a perisomatic shunting inhibition driven by SuM in area CA2. Furthermore, we found that with elevated cholinergic tone, recruitment of BCs by SuM controlled PNs excitability and shaped spontaneous burst firing. This finding demonstrates that SuM activity can pace action potential firing in PNs through recruitment of feedforward inhibition.

The population of INs potently excited by SuM transmission display many features that motivate us to classify them as PV+ BCs. They have somas located in the somatic layer, have densely packed perisomatic-targeted axons, are fast spiking and show PV immuno-reactivity. The selective expression of GiDREADD in PV+ cells allows for selective silencing that reduces SuM-driven feedforward inhibition of area CA2 PNs. With these techniques, however, we were unable to sufficiently silent PV+ cells in area CA2, leaving open the possibility that another population of basket cell is targeted by SuM input. The feedforward inhibitory transmission recruited by SuM stimulation is highly sensitive to MOR activation. While this supports our hypothesis that PV+ cells are targeted by SuM input, MORs are not entirely exclusive to PV+ cells (*Stumm et al., 2004*). We also show that the SuM-recruited feedforward inhibition is sensitive to DOR activation. Unlike MORs, DORs have been shown to be specific to PV+ cells in area CA2, however, only a sub-population of PV +INs express this

receptor (*Nasrallah et al., 2019*; *Piskorowski and Chevaleyre, 2013*) leaving open the possibility that the remaining IPSCs evoked by SuM stimulation are not from PV+ cells. We also show in this work that SuM-evoked inhibitory currents are blocked by the application of ω-agatoxin TK, indicating that these recruited INs express P/Q-type CaV channels, consistent with PV+ BCs (*Zaitsev et al., 2007*). However, we also saw that ω-agatoxin TK also blocked gluatamatergic transmission from SuM inputs, preventing a simple interpretation of these results. Thus, while there is ample evidence that SuM inputs target PV+ BCs in area CA2, we cannot exclude the possibility that other populations of BCs, such as CCK+ INs are also targeted by these inputs. PV+ BCs in the hippocampus have been shown to be modulated by CCK (*Lee et al., 2011*) which would have very interesting implications for the effect of SuM activity in area CA2. Furthermore, it was recently shown that PV+ BCs actively inhibit CCK+ BCs, enabling a complementary perisomatic inhibitory system that allows for brain-state-dependent activity during behavior (*Dudok et al., 2021*).

Recent studies have indicated that the SuM input to CA2 plays a key role in social novelty discrimination (*Chen et al., 2020*). Our findings are very consistent with the finding that DOR-mediated inhibitory synaptic plasticity of PV+ INs in area CA2 is required for social recognition memory (*Domínguez et al., 2019*). Furthermore, exposure to a novel conspecific induces a DOR-mediated plasticity in this same inhibitory network in area CA2 (*Leroy et al., 2017*). Thus, our finding that SuM input acts via PV+ interneurons fits with previous studies and provides a link between social novelty detection and local CA2 hippocampal inhibitory plasticity.

Overall, the local circuitry and consequences of SuM input to area CA2 contrasts with the SuM-DG path (*Hashimotodani et al., 2018*; *Li et al., 2020*; *Mizumori et al., 1989*; *Nakanishi et al., 2001*). Previously, we have shown that unlike the SuM-DG synapse, the SuM-CA2 synapse is entirely glutamatergic (*Chen et al., 2020*). In this study, we use both a VGluT2-Cre and SuM-Cre mouse lines to demonstrate how the combination of direct excitation and feedforward inhibition regulates CA2 PN AP firing. Our data shows that SuM activity results in synchronized feedforward inhibition from CA2 to CA1 which decreases CA1 PN firing. While our results are very intriguing given the importance of area CA2 in propagation of hippocampal network activity (*Oliva et al., 2016a*), further questions remain. CA2 PNs also receive excitatory input from DG cells via the mossy fibers (*Kohara et al., 2014*; *Llorens-Martín et al., 2015*). It has been postulated that by increasing DG excitability, the SuM may also be indirectly acting on CA2 (*Silkis and Markevich, 2020*). These circuits merit further exploration.

## Consequences of SuM input on area CA2 output

Recent work has demonstrated a strong excitatory drive from area CA2 to CA1 (*Chevaleyre and Siegelbaum, 2010*; *Kohara et al., 2014*; *Nasrallah et al., 2019*). Consequently, modification of CA2 output through synaptic plasticity (*Nasrallah et al., 2019*) or neuromodulation (*Tirko et al., 2018*) affects CA1 activity. This observation is critical when considering social memory formation, which is known to depend on CA2 output (*Hitti and Siegelbaum, 2014*; *Stevenson and Caldwell, 2014*) and is likely encoded in downstream ventral CA1 (*Okuyama et al., 2016*). CA2-targeting cells in the SuM have recently been shown to be highly active during novel social exploration (*Chen et al., 2020*). From our results, we hypothesize that this novel social signal from the SuM, acts via the PV +inhibitory network in area CA2 to control the timing of CA2 output onto area CA1.

By recruiting feedforward inhibition, SuM activity paces and temporally constrains AP firing from CA2 PNs undergoing depolarization. More critically, in conditions of elevated cholinergic tone relevant to SuM activity in vivo, CA2 PNs depolarize and fire bursts of APs that can be shaped by SuM input both by controlling AP firing as well as membrane depolarization. While this result was obtained by triggering SuM input stimulation to the onset of burst firing by CA2 PNs, in vivo and acute slice experiments revealed a consistent influence of CA1 PN AP firing by SuM input to area CA2 regardless of the timing of SuM input stimulation relative to CA2 PN AP burst firing. These results demonstrate a powerful control of SuM input over CA2 output when PNs are spontaneously firing bursts of APs, a firing mode that is most efficient at influencing CA1 activity (*Tirko et al., 2018*). Optogenetic experiments have recently shown that CA2 PNs can drive a strong feedforward inhibition in area CA1 (*Kohara et al., 2014*; *Nasrallah et al., 2019*). Although SuM input likely does not directly drive feedforward inhibition in area CA1 (*Chen et al., 2020*), the recruitment of feedforward inhibition in area CA2 by SuM input activation could curtail the time window of spontaneous firing in CA2 PNs and effectively lead to a synchronized drive of feedforward inhibition by area CA2

over area CA1. We postulate that the concerted IPSC that we detect in area CA1 with SuM fiber photostimulation in area CA2 corresponds to the large decrease in firing that is observed in CA1 multi-unit recordings in vivo. Thus, these data provide evidence for a long-range control of CA2 bursting activity and the consequences in downstream area CA1 in conditions of high cholinergic tone that accompanies theta oscillations in vivo during which SuM is active.

## Relevance of the SuM input to area CA2 for hippocampal oscillations

The activity of hippocampal neurons is orchestrated by brain rhythms, notably theta and gamma oscillations that are prominent during exploration and linked to the learning and memory functions of the hippocampus (*Buzsáki, 2002*; *Buzsáki and Wang, 2012*; *Colgin, 2016*). The SuM is active during these brain states and contributes to theta oscillations in the hippocampus (*Kirk et al., 1996*; *Kirk and McNaughton, 1993*; *Kocsis and Vertes, 1994*; *McNaughton et al., 1995*; *Pan and McNaughton, 2002*; *Pan and McNaughton, 1997*; *Thinschmidt et al., 1995*). Here, we show that the SuM controls area CA2 output to CA1 by recruiting PV +BCs, which are important for both theta and gamma oscillations (*Fuchs et al., 2007*; *Gulyás et al., 2010*; *Korotkova et al., 2010*; *Mann and Mody, 2010*). Through its perisomatic mono-synaptic excitation and PV +BC-mediated di-synaptic inhibition of CA2 PNs, the SuM likely contributes to enforcing theta-locked windows of excitability shaping CA2 PNs output. Area CA2 can influence CA1 activity not only by direct projections but also through its interactions with both CA3 (*Boehringer et al., 2017*) and EC (*Chevaleyre and Siegelbaum, 2010*; *Rowland et al., 2013*) which are major contributors to CA1 theta and gamma oscillations (*Buzsáki, 2002*; *Colgin, 2016*). CA2 axons target both CA1 *stratum oriens* and *radiatum* (*Nasrallah et al., 2019*), thus the CA2 projections to CA1 likely contributes to the theta and slow gamma oscillations observed in these strata in CA1 (*Belluscio et al., 2012*; *Colgin et al., 2009*; *Schomburg et al., 2014*). Indeed, CA2 PNs show theta- and gamma-modulation of their activity (*Fernandez-Lamo et al., 2019*; *Oliva et al., 2016b*), and chemogenetic manipulations of their excitability bidirectionally influences hippocampal low gamma power (*Alexander et al., 2018*). Further, chronic block of CA2 output transmission leads to hippocampal hyperexcitability and disrupts CA1 theta phase preference and spatial coding (*Boehringer et al., 2017*). Therefore, by providing a theta-locked input shaping CA2 PN activity, the SuM is poised to contribute to oscillatory activity in downstream brain regions receiving CA2 input. Indeed, chemogenetic activation or silencing of SuM glutamatergic neurons respectively increases or decreases theta and gamma power in the EEG (*Pedersen et al., 2017*). Further, the SuM is involved in coordinating activity between the prefrontal cortex, the thalamus and area CA1 as evidenced by a loss of theta coherence amongst these regions upon SuM optogenetic silencing during a spatial task requiring action planning (*Ito et al., 2018*). Altogether, these studies point to the SuM as a crucial component in the regulation of hippocampal oscillations and our findings shed light on an aspect of this circuit.

## Gating of area CA2 activity by PV +INs and significance for pathologies

The density of PV +INs in area CA2 is strikingly higher than in neighboring areas CA3 and CA1 (*Botcher et al., 2014*; *Piskorowski and Chevaleyre, 2013*). This population of INs has been shown to play a powerful role in controlling the activation of CA2 PNs by CA3 inputs (*Nasrallah et al., 2015*). We show in this study that long-range inputs from the SuM can strongly recruit PV +BCs, which in turn inhibit PNs in this area. Hence, both intra-hippocampal inputs from CA3 and long-range inputs from the SuM converge onto PV +INs to control CA2 PN excitability and output.

Postmortem studies have reported losses of PV +INs in area CA2 in pathological contexts including bipolar disorder (*Benes et al., 1998*), Alzheimer's disease (*Brady and Mufson, 1997*), and schizophrenia (*Benes et al., 1998*; *Knable et al., 2004*). Consistent with these reports, in a mouse model of the 22q11.2 deletion syndrome, a major risk factor for schizophrenia in humans, we found a loss of PV staining and deficit of inhibitory transmission in area CA2 that were accompanied by impairments in social memory (*Piskorowski et al., 2016*). We postulate that the PV +INs altered during pathological conditions may be the same population of PV +BCs recruited by long-range SuM inputs. Indeed, the DOR-mediated plasticity onto PV +INs is altered in the 22q11.2 deletion syndrome mouse model, and we show here that a fraction of the PV +INs targeted by the SuM also express DOR. Thus, the loss of function of PV +INs in area CA2 could disrupt proper long-range connection between the hippocampus and the hypothalamus and possibly contribute to some of the

cognitive impairments observed in schizophrenic patients and animal models. Further, pharmacological mouse models of schizophrenia have reported increased c-fos immunoreactivity in the SuM as well as memory impairments (*Castañé et al., 2015*). Although several alterations in these models of schizophrenia could lead to deficits of hippocampal-dependent behavior, abnormalities of the SuM projection onto area CA2 appear as a potential mechanism that warrants further investigation.

# Materials and methods

### Key resources table

| Reagent type (species) or resource | Designation | Source or reference | Identifiers | Additional information |
|---|---|---|---|---|
| Genetic reagent (*Mus. musculus*) | Tg(Slc17a-icre)10Ki | **Borgius et al., 2010** | Tg(Slc17a-icre)10Ki; VGluT2-cre | |
| Genetic reagent (*Mus. musculus*) | Csf2rb2-Cre | **Chen et al., 2020** | Csf2rb2-Cre; SuM-cre | |
| Genetic reagent (*Mus. musculus*) | Pvalbtm1(cre)Arbr/J (PV-Cre) | Jackson | RRID:IMSR_JAX:017320 | |
| Genetic reagent (*adeno-associated virus*) | AAV9.EF1a.DIO. hChR2(H134R).EYFP | Addgene | RRID:Addgene 20298 | |
| Genetic reagent (adeno-associated virus) | AAV9.hSynapsin. EGFP.WPRE.bGH | Addgene | RRID:Addgene 51502 | |
| Genetic reagent (adeno-associated virus) | AAV.Synapsin. DIO.hM4D(Gi). mCherry | McHugh Laboratory, Riken | | |
| Genetic reagent (adeno-associated virus) | AAV2/9.hSyn. hChR2 (H134R).EYFP. WPRE.hGH | Addgene | RRID:Addgene 26973 | |
| Genetic reagent (Canine adeno virus) | CAV2-cre | Platforme de Vectorologie de Montpellier | CAV Cre | |
| Antibody | Anti-RGS14 (mouse monoclonal) | NeuroMab | 73–422 RRID:AB_2877596 | (1:300) |
| Antibody | Anti-GFP (chicken polyclonal) | Abcam | ab13970 RRID:AB_300798 | (1:10,000) |
| Antibody | Anti-VGluT2 (guinea pig polyclonal) | Millipore | AB2251 RRID:AB_1587626 | (1:10000) |
| Antibody | Anti-parvalbumin (rabbit polyclonal) | Swant | PV27 RRID:AB_2631173 | (1:2000) |
| Antibody | Anti- PCP4 (rabbit polyclonal) | Sigma | HPA005792 RRID:AB_1855086 | (1:600) |
| Antibody | Anti-Calretinin (mouse monoclonal) | Millipore | MAB1568 RRID:AB_94259 | (1:500) |
| Antibody | Anti-mCherry (rat monoclonal) | Life technologies | M11217 RRID:AB_2536611 | (1:5000) |
| Other | Far-red neurotrace | Life technologies | N21483 RRID:AB_2572212 | (1:300) |
| Peptide, recombinant protein | Alexa-546-conjugated streptavidin | Life Technologies | S11225 RRID:AB_2532130 | (1:500) |
| Peptide, recombinant protein | Biocytin | HelloBio | HB5035 | (4mg/mL) |

*Continued on next page*

*Continued*

| Reagent type (species) or resource | Designation | Source or reference | Identifiers | Additional information |
|---|---|---|---|---|
| Chemical compound, drug | NBQX | HelloBio | HB0443 | (10 µM) |
| Chemical compound, drug | D-APV | HelloBio | HB0225 | (50 µM) |
| Chemical compound, drug | SR95531 | Tocris | 1262 | (1 µM) |
| Chemical compound, drug | CGP55845A | Tocris | 1248 | (2 µM) |
| Chemical compound, drug | DPDPE | Alfa Aesar | J66293 | (0.5 µM) |
| Chemical compound, drug | DAMGO | Tocris | 1171 | (1 µM) |
| Chemical compound, drug | Clozapine N-oxide (CNO) | HelloBio | HB1807 | (10 µM) |
| Chemical compound, drug | Tetrodotoxin (TTX) | Tocris | 1078 | (0.2 µM) |
| Chemical compound, drug | Carbamoylcholine chloride (CCh) | Tocris | 2810 | (10 µM) |
| Chemical compound, drug | ω-agatoxin TK | Alomone labs | STA-530 | (200 nM) |
| Software, algorithm | Matlab | Mathworks | http://www.mathworks.com RRID:SCR_001622 | |
| Software, algorithm | Igor Pro | Wavemetrics | http://www.wavemetrics.com RRID:SCR_000325 RRID:SCR_004186 (neuromatic) | |
| Software, algorithm | OriginPro | OriginLab Corporation | http://www.originlab.com RRID:SCR_014212 | |
| Software, algorithm | pClamp | Molecular Devices | http://www.moleculardevices.com RRID:SCR_011323 | |
| Software, algorithm | Axograph | Axograph | http://www.axograph.com RRID:SCR_014284 | |

All procedures involving animals were performed in accordance with institutional regulations (French Ministry of Research and Education protocol #12406–2016040417305913). Animal sample sizes were estimated using power tests with standard deviations and ANOVA values from pilot experiments. A 15% failure rate was assumed to account for stereotaxic injection errors and slice preparation complications. Every effort was made to reduce animal suffering.

### Use of the *Tg(Slc17a-icre)10Ki* mouse line

We used the Tg(Slc17a-icre)10Ki mouse line that was previously generated (*Borgius et al., 2010*) and expresses the Cre recombinase under control of the *slc17a6* gene coding the vesicular glutamate transporter isoform 2 (VGluT2).

### Use of the *Csf2rb2-Cre* mouse line

We used the *Csf2rb2-Cre* mouse line that was recently generated (*Chen et al., 2020*) and expresses the Cre recombinase under control of the *Csf2rb2* gene that shows selective expression in the SuM.

### Use of the *Pvalbtm1(cre)Arbr/J* mouse line

We used the Pvalbtm1(cre)Arbr/J mouse line that was previously generated (*Hippenmeyer et al., 2005*) and expresses the Cre recombinase under control of the *Pvalbm* gene coding parvalbumin (PV).

### Stereotaxic viral injection

Animals were anaestetized with ketamine (100 mg/kg) and xylazine (7 mg/kg). The adeno-associated viruses AAV9.EF1a.DIO.hChR2(H134R).EYFP and AAV9.hSynapsin.EGFP.WPRE.bGH were used at $3 \times 10^8$ vg, the AAV.Synapsin.DIO.hM4D(Gi).mCherry was used at $3.6 \times 10^9$ vg and the AAV2/9.hSyn.hChR2(H134R).EYFP.WPRE.hGH was used at $3.7 \times 10^{13}$ vg. The retrograde tracer CAV2-cre virus was used at $2.5 \times 10^{12}$ vg. 500 nL of virus was unilaterally injected into the brain of 4-week-old male wild type C57BL/6, Tg(Slc17ab-icre)10Ki (VGluT2-Cre), csf2rb2-cre (SuM-Cre) or Pvalbtm1(cre)Arbr/J (PV-Cre) mice at 100 nL/min and the injection cannula was left at the injection site for 10 min following infusion. In the case of AAV.Synapsin.DIO.hM4D(Gi)-mCherry injection in PV-Cre mice, bilateral injections were performed in dorsal CA2. The loci of the injection sites were as follows: anterior–posterior relative to bregma: −2.8 mm for SuM, −1.6 mm for CA2; medial-lateral relative to midline: 0 mm for SuM, 1.9 mm for CA2; dorsal-ventral relative to surface of the brain: 4.75 mm for SuM, 1.4 mm for CA2.

### Electrophysiological recordings

Transverse hippocampal slices were prepared at least 3 weeks after viral injection and whole-cell patch-clamp recordings were performed from PNs and INs across the hippocampal CA regions. In the case of PV-Cre mice injected with AAV.Synapsin.DIO.hM4D(Gi)-mCherry, slices were prepared 6 weeks after viral injection. Animals were deeply anesthetized with ketamine (100 mg/kg) and xylazine (7 mg/kg), and perfused transcardially with a N-methyl-D-glucamin-based (NMDG) cutting solution containing the following (in mM): NMDG 93, KCl 2.5, NaH$_2$PO$_4$ 1.25, NaHCO$_3$ 30, HEPES 20, glucose 25, thiourea 2, Na-ascorbate 5, Na-pyruvate 3, CaCl$_2$ 0.5, MgCl$_2$ 10. Brains were then rapidly removed, hippocampi were dissected out and placed upright into an agar mold and cut into 400-μm-thick transverse slices (Leica VT1200S) in the same cutting solution at 4℃. Slices were transferred to an immersed-type chamber and maintained in artificial cerebro-spinal fluid (ACSF) containing the following (in mM): NaCl 125, KCl 2.5, NaH$_2$PO$_4$ 1.25, NaHCO$_3$ 26, glucose 10, Na-pyruvate 2, CaCl$_2$ 2, MgCl$_2$ 1. Slices were incubated at 32℃ for approximately 20 min then maintained at room temperature for at least 45 min prior to patch-clamp recordings performed with either potassium- or cesium-based intracellular solutions containing the following (in mM): K- or Cs-methyl sulfonate 135, KCl 5, EGTA-KOH 0.1, HEPES 10, NaCl 2, MgATP 5, Na$_2$GTP 0.4, Na$_2$-phosphocreatine 10 and biocytin (4 mg/mL).

ChR2 was excited by 488 nm light delivered by a LED attached to the epifluorescence port of the microscope. Light stimulations trains consisted of 2–10 pulses, 0.5 ms long, delivered at 10 Hz, repeated every 20 s for at least 20 sweeps. Stimulating pipettes filled with ACSF were placed in *stratum radiatum* (SR) of CA1 to antidromically excite CA3-CA2 synapses and in *stratum lacunosum moleculare* (SLM) to stimulate distal dendritic inputs in area CA2. Synaptic currents were evoked with a constant voltage stimulating unit (Digitimer Ltd.) set at 0.1 ms at a voltage range of 5–10 V. For the patch-clamp recordings in area CA1 with stimulation of SuM axons in area CA2, 50 ms long light stimulation pulses were delivered every 10 s. We used a light intensity of 25 mW/mm$^2$ which was experimentally determined as the lowest irradiance allowing TTX-sensitive maximal responses in all cell types and conditions. Data were obtained using a Multiclamp 700B amplifier, sampled at 10

kHz and digitized using a Digidata. The pClamp10 software was used for data acquisition. Series resistance were <20 MOhm and were not compensated in voltage-clamp, bridge balance was applied in current-clamp. An experimentally determined liquid junction potential of approximately 9 mV was not corrected for. Pharmacological agents were added to ACSF at the following concentrations (in µM): 10 NBQX and 50 D-APV to block AMPA and NMDA receptors, 1 SR95531 and 2 CGP55845A to block GABA$_A$ and GABA$_B$ receptors, 1 DAMGO to activate µ-opioid receptors (MOR), 0.5 DPDPE to activate δ-opioid receptors (DOR), 10 clozapine N-oxide (CNO) to activate hM4D(Gi) DREADDs, 10 CCh to activate cholinergic receptors, 0.2 tetrodotoxin (TTX) to prevent sodic action potential generation, 200 nM ω-agatoxin TK to block P/Q-type voltage-gated calcium channels.

## Surgery for in vivo recordings

All surgeries were performed in a stereotaxic frame (Narishige). Csf2rb2-cre male mice from 3 to 6 months of age were anaesthetized using 500 mg/kg Avertin. pAAV.DIO.hChR2(H134R).EYFP was injected into the SuM (−2.7 mm AP, +0.4 mm ML, −5.0 mm DV) using a 10 µL Hamilton microsyringe (701LT, Hamilton) with a beveled 33 gauge needle (NF33BL, World Precision Instruments (WPI)). A microsyringe pump (UMP3, WPI) with controller (Micro4, WPI) were used to set the speed of the injection (100 nl/min). The needle was slowly lowered to the target site and remained in place for 5 min prior to start of the injection and the needle was removed 10 min after infusion was complete. Following virus injection, a custom-built screw-driven microdrive containing six independently adjustable nichrome tetrodes (14 µm diameter), gold-plated to an impedance of 200–250 kΩ was implanted, with a subset of tetrodes targeting CA1, and an optic fiber (200 µm core diameter, NA = 0.22) targeting CA2 (−1.9 mm AP, ±2.2 mm ML, −1.6 mm DV). Following recovery, the tetrodes were slowly lowered over several days to CA1 pyramidal cell layer, identified by characteristic local field potential patterns (theta and sharp-wave ripples) and high amplitude multiunit activity. During the adjustment period the animal was habituated every day to a small box in which recording and stimulation were performed.

## In vivo recording protocol

Recording was commenced following tetrodes reaching CA1. To examine the impact of SuM terminal stimulation in CA2 the mice were returned to the small familiar box and trains of 10 light pulses (473 nm, 10 mW/mm$^2$ and pulse width 50 ms) were delivered to the CA2 at 10 Hz. The pulse train was repeated every 10 s for at least 20 times as the animals freely explored the box. Multiunit activity was recorded using a DigitalLynx 4SX recording system running Cheetah v.5.6.0 acquisition software (Neuralynx). Broadband signals from each tetrode were filtered between 600 and 6000 Hz and recorded continuously at 32 kHz. Recording sites were later verified histologically with electrolytic lesions as described above and the position of the optic fiber was also verified from the track.

## In vivo data analysis

Spike and event timestamps corresponding to onset of each laser pulse were imported into Matlab (MathWorks) and spikes which occurred 50 ms before and 100 ms after each laser pulse were extracted. Raster plots were generated using a 1 ms bin size. Similar results were obtained using 5 ms and 10 ms bin size (data not shown). Firing rate histograms were calculated by dividing total number of spikes in each time bin by that bin's duration. Each firing rate histogram was normalized by converting it into z-score values. Mean standard deviation values for the z-score calculation were taken from pre-laser pulse time period. To average the response across all mice, for each tetrode the firing rate in each bin was normalized to the average rate in the pre-laser period.

## Immunochemistry and cell identification

Midbrains containing the injection site were examined post-hoc to ensure that infection was restricted to the SuM.

Post-hoc reconstruction of neuronal morphology and SuM axonal projections were performed on slices and midbrain tissue following overnight incubation in 4% paraformaldehyde in phosphate buffered saline (PBS). Midbrain sections were re-sliced sagittally to 100 µm thick sections. Slices were permeabilized with 0.2% triton in PBS and blocked overnight with 3% goat serum in PBS with 0.2%

triton. Primary antibody (Life technologies) incubation was carried out in 3% goat serum in PBS overnight at 4°C. Channelrhodopsin-2 was detected by chicken primary antibody to GFP (Life technologies) (1:10,000 dilution) and an alexa488-conjugated goat-anti chick secondary. Other primary antibodies used were mouse anti-RGS14 (Neuromab) (1:300 dilution), rabbit anti- PCP4 (Sigma) (1:600 dilution), guinea pig anti-VGluT2 antibody (Milipore) (1:10,000 dilution), rabbit anti-parvalbumin antibody (Swant) (1:2000 dilution). Alexa-546-conjugated streptavidin (Life technologies), secondary antibodies and far-red neurotrace (Life technologies) incubations were carried out in block solution for 4 hr at room temperature. Images were collected with a Zeiss 710 laser-scanning confocal microscope.

Reconstructed neurons were classified as either PNs or INs based on the extension and localization of their dendrites and axons. PNs were classified as deep (closest to *stratum oriens*) or superficial (closest to *stratum radiatum*) based on the radial position of their soma in the pyramidal layer. CA1, CA2, and CA3 PNs were identified based on their somatic localization, dendritic arborization and presence of thorny excrescences (TE). Among INs with somas located in the pyramidal layer (*stratum pyramidale*, SP), discrimination between BCs and non-BCs was achieved based on the restriction of their axons to SP or not, respectively. When available, firing patterns upon injection of depolarizing current step injection, action potential (AP) half-width, amount of repolarizing sag current upon hyperpolarization from $-70$ mV to $-100$ mV by current step injection, membrane resistance ($R_M$) and capacitance ($C_M$) were additionally used for cell identification. CA2 and CA3a PNs as well as superficial and deep PNs displayed similar firing patterns, AP width, sag current, $R_M$ and $C_M$; the only statistically difference being a larger $R_M$ of CA3a compared to CA2 PNs which is consistent with previous studies (*Chevaleyre and Siegelbaum, 2010*; *Sun et al., 2017*). In contrast, INs had faster firing rates, shorter AP width, higher $R_M$ and lower $C_M$ than PNs. BCs further differed from non-BCs by the presence of a larger sag potential. All recorded neurons that could not be unequivocally identified as PNs or INs were excluded from analysis. SuM connectivity to each neuronal population was quantified by dividing the number of cells that displayed a post-synaptic response to SuM input stimulation by the total number of cells sampled for each neuronal population across all recording sessions with successful SuM-CA2 transmission.

## Data analysis and statistics

Electrophysiological recordings were analyzed using IGORpro (Wavemetrics) and Clampfit (Molecular devices) software. For accurate measurements of the kinetics and latencies of post-synaptic responses, the following detection process was used. For each cell, average traces were used to create a template waveform that was then fitted to individual traces and measurements were performed on the fitted traces. When only amplitudes of responses were needed, traces were baselined and amplitudes were simply measured at the peak of the responses. Firing adaptation indexes were computed as the ratio of the initial firing rate (instantaneous firing frequency of the first spikes) over the steady-state firing rate (average instantaneous firing frequency for spikes occurring after 500 ms in the 1 second-long current step). Results are reported ± SEM. Statistical significance was assessed using $\chi^2$ test, Student's T test, Mann-Whitney U test, Wilcoxon signed-rank test, Kruskal-Wallis test, one-way or two-way ANOVA where appropriate.

## Acknowledgements

Equipment for the IPNP mouse husbandry facility was funded by the *Région Ile de France*. This work was supported by the RIKEN Center for Brain Science (TJM), Grant-in-Aid for Scientific Research from MEXT (19H05646; TJM), Grant-in-Aid for Scientific Research on Innovative Areas from MEXT (19H05233; TJM), ANR-13-JSV4-0002-01 (RAP), ANR-18-CE37-0020-01 (RAP), the Ville de Paris Programme Emergences (RAP), and the Brain and Behavioral Research Foundation NARSAD Young Investigator Grant (RAP) and the Foundation Recherche Médicale, FRM:FTD20170437387 and a gift from Schizo-Oui (VR).

## Additional information

### Funding

| Funder | Grant reference number | Author |
|---|---|---|
| RIKEN Brain Science Institute | | Thomas J McHugh |
| Ministry of Education, Culture, Sports, Science and Technology | 19H05646 | Thomas J McHugh |
| Ministry of Education, Culture, Sports, Science and Technology | 19H05233 | Thomas J McHugh |
| Agence Nationale de la Recherche | ANR-13-JSV4-0002-01 | Rebecca Ann Piskorowski |
| Agence Nationale de la Recherche | ANR-18-CE37-0020-01 | Rebecca Ann Piskorowski |
| Ville de Paris | Programme Emergences | Rebecca Ann Piskorowski |
| Brain and Behavior Research Foundation | NARSAD Young INvestigator Award | Rebecca Ann Piskorowski |
| Fondation pour la Recherche Médicale | FTD20170437387 | Vincent Robert |
| Schizo-Oui | | Vincent Robert |

The funders had no role in study design, data collection and interpretation, or the decision to submit the work for publication.

### Author contributions

Vincent Robert, Conceptualization, Resources, Formal analysis, Investigation, Writing - original draft, Writing - review and editing; Ludivine Therreau, Roman Boehringer, Formal analysis, Investigation; Vivien Chevaleyre, Eude Lepicard, Investigation, Writing - review and editing; Cécile Viollet, Project administration, Writing - review and editing; Julie Cognet, Resources, Investigation, Project administration; Arthur JY Huang, Resources, Methodology; Denis Polygalov, Formal analysis, Writing - review and editing; Thomas J McHugh, Supervision, Funding acquisition, Investigation, Writing - review and editing; Rebecca Ann Piskorowski, Conceptualization, Resources, Formal analysis, Supervision, Funding acquisition, Validation, Methodology, Writing - original draft, Project administration, Writing - review and editing

### Author ORCIDs

Roman Boehringer http://orcid.org/0000-0003-2856-3262
Denis Polygalov http://orcid.org/0000-0002-8165-5257
Thomas J McHugh http://orcid.org/0000-0002-1243-5189
Rebecca Ann Piskorowski https://orcid.org/0000-0003-0120-2360

### Ethics

Animal experimentation: All procedures involving animals were performed in accordance with institutional regulations (French Ministry of Research and Education protocol #12406-2016040417305913). Animal sample sizes were estimated using power tests with standard deviations and ANOVA values from pilot experiments. A 15% failure rate was assumed to account for stereotaxic injection errors and slice preparation complications. Every effort was made to reduce animal suffering.

### Decision letter and Author response

Decision letter https://doi.org/10.7554/eLife.63352.sa1
Author response https://doi.org/10.7554/eLife.63352.sa2

## Additional files

### Supplementary files
• Transparent reporting form

### Data availability
The data analysed for this study are included in the manuscript and supporting files. These are included in tables, and data file for Figure 7.

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
