## [Decision Letter]

**Acceptance summary:**

The study describes the properties of inputs from the supramammillary nucleus (SuM) to the CA2 area of the hippocampus. This is an important area of study given the increasing importance of SuM inputs and area CA2 in social memory encoding.

Novel information is presented in this paper about the influence of the SuM input on the local hippocampal network in CA2 and the effects of this input on network activity in CA1. The authors use complimentary methods to address this question including patch-clamp recordings and optogenetics. The study will be of interest to readers who study hippocampal function, supramammillary nucleus function, social memory, or neuropsychiatric disorders in which CA2 circuits are affected.

**Decision letter after peer review:**

Thank you for submitting your article "Local circuit allowing hypothalamic control of hippocampal area CA2 1 activity and consequences for CA1" for consideration by *eLife*. Your article has been reviewed by 3 peer reviewers, one of whom is a member of our Board of Reviewing Editors, and the evaluation has been overseen by Laura Colgin as the Senior Editor. The following individual involved in review of your submission has agreed to reveal their identity: Sreedharan Sajikumar (Reviewer #2).

The reviewers have discussed the reviews with one another and the Reviewing Editor has drafted this decision to help you prepare a revised submission.

The study describes the properties of inputs from the supramammillary nucleus (SuM) to the CA2 area of the hippocampus. Novel information is presented by the influence of the SuM input on the local hippocampal network in the CA2 and what the effect of this input is on network activity in the CA1. The authors use complimentary methods to address this question including patch-clamp recordings and optogenetics. Overall the reviewer found this study important, the experiments well-designed and the data of high quality. However, there are several key points raised by the reviewers to strengthen the data in order to fully support the authors' conclusions, addressing these will require additional experimental work. The list below summarizes the list of required experiments reviewers agreed would be necessary for having full confidence in the authors' conclusions. Reviewers' individual comments contain additional issues that would require the authors to make changes to the discussion, figure legends etc.

1. The authors would need to show the effect of SuM stimulation on on synaptically triggered APs and not only on Aps evoked with a current step.

2. The change in the balance of EPCs and IPSCs in a train should be demonstrated in a single cell.

3. The properties of monosynaptic/disynaptic events should be compared and the lack of direct GABAergic input from the SuM demonstrated. The authors should quantify the delay time to light-evoked IPSCs to address whether the SuM-CA2 inputs are forming monosynaptic or disynaptic GABAergic connections to pyramidal neurons, as it is possible SuM neurons co-release glutamate and GABA to CA2. Given the importance of the mono vs. disynaptic innervation of different types of cells, the authors should go beyond the TTX experiments (as TTX would block a disynaptic EPSC) and also use 4-AP to recover the TTX blocked current to unequivocally prove that they inputs are monosynaptic.

4. The preferential role of PV+ cells should be shown with a more selective pharmacological approach.

5. The authors should elaborate on how SuM stimulation influences theta/γ rhythms in the CA1 area.

*Reviewer #1:*

In this study Robert et al. describes the properties of long-range projections from the SuM to the CA2 area of the hippocampus. The authors identified direct excitatory and indirect inhibitory drive from SuM inputs on CA2 pyramidal neurons and showed that direct excitatory drive impinges on PV-positive basket cells. The overall effect of the input on CA2 activity was an increased precision of APs. The study also suggests that the input from the CA2 drives inhibition in the CA1 area. The study provides very interesting and new information about the cellular properties of SuM input in the CA2 area. This is an important question given the increasing importance of SuM inputs in social memory encoding. The study is timely, currently we have very limited data about the features and exact cellular profile of this input. The study is using elegant technical approaches o answer the central question of the study. While the study is addressing an important question and provides novel data, the authors central claim about the role of feedforward inhibition would need to be strengthened by the addition of experiments addressing how E-I balance changes in trains in individual neurons and how this can be linked to changes in the temporal precision of synaptically evoked APs.

Action potentials are evoked with a current step. Since the study is focused on the network effects of feedforward inhibition, it would be useful to see how the properties of synaptically evoked action potentials change. In the cortex and in the CA1 feed forward inhibition was shown to limit the temporal summation of excitatory inputs which lead to decrease in AP jitter (Gabernet et al., 2005, Pouille and Scanziani, 2001). In order to map these dynamics APs should be evoked via synaptic stimulation and not through current injection.

The authors show recordings of monosynaptic EPSCs in pyramidal cells and interneurons. It would be important to know how inhibitory and excitatory PSCs change in a train. Recordings from single cells held at E-GLUT ad E-GABA would allow the authors to monitor excitatory and inhibitory events in a train ad map how their balance changes. Can the change in E-I balance explain the change in AP jitter?

What are the characteristics of the SuM-driven inhibitory currents? Does the latency and jitter of monosynaptic EPSCs and disynaptic IPSCs differ? If one is monosynaptic and the other is disynaptic one would expect significant differences in both of these parameters.

How do the authors exclude the contribution of feed-back inhibition? feedforward and feed-back inhibition both could have an impact on the temporal precision of APs.

*Reviewer #2:*

The article brings to light the functional consequences of the activity of SuM afferents terminating at CA2 neurons in the hippocampus using a combination of a variety of methods like whole-cell voltage clamp and optogenetics. In addition, the authors provide evidence that modulation of the CA2 neurons by SuM afferents affects the activity pattern of CA1 neurons. Specifically, the study reveals that the 'functional' connectivity between SuM and CA2 is mainly mediated by the regulation of PV+ basket cells that are involved in the feed forward inhibition of CA2 principal neurons. This study is also relevant in the context of neuropsychiatric disorders where PV+ IN density in the CA2 area is preferentially reduced.

It would be good if some results and implications are further clarified for better understanding in the Discussion section:

1. The results indicate that SuM recruits a feed forward inhibition onto CA2 PNs, which contributes to the shaping of CA2 AP firing. However, it is not entirely intuitive how the feed forward inhibition of CA2 PNs by SuM also reduces CA1 activity, as CA2 has also been known to recruit strong feed forward inhibition onto CA1. This would intuitively suggest that decrease in CA2 activity by photostimulation of SuM afferents will in turn decrease the feed forward inhibition by CA2 onto CA1, and thereby increase CA1 activity. However, the results suggest otherwise. Would this be suggestive of a stronger direct excitatory projection from CA2 to CA1 PNs that is more dominant than the feed forward inhibition of CA1 PNs by CA2? This may be a good point to further elaborate on in the Discussion section, so that the effect of SuM-CA2 connectivity on CA1 output becomes clearer.

2. In the introduction section line 44, it is written that 'CA2 neurons do not undergo NMDA-mediated synaptic plasticity'. This may not be always the case; rather it may be better to rephrase 'NMDA-mediated' as 'high frequency stimulation-induced'. It has been shown previously that NK1 receptor activation by pharmacological application of substance P in hippocampal slices triggers a slow onset NMDA-dependent LTP in CA2 neurons by high frequency stimulation of CA3 afferents to CA2 (Dasgupta et al., 2017).

3. Line 250: "BC transmission is insensitive to MOR activation (Glickfeld et al., 2008)."

Was the Glickfeld study done in CA2 neurons? If not, it would be good to show that PV+ CA2 BCs are also sensitive to DAMGO and to what degree?

The experiment shows that IPSC in PNs are inhibited by DAMGO that should have enhanced light induced EPSCs if PV+ BCs are responsible for feed forward inhibition. But it seems that has not been observed. What are direct EPSCs – electrical stimulation of CA3-CA2 synapses?

4. Overall, the results seem to suggest that SuM stimulation would induce a net inhibition (?) of CA2 PNs by recruiting interneurons (INs). However, the role played by the direct glutamatergic connections from SuM to CA2 PNs is not entirely clear. Is it less prominent due to sparse SuM-PN projections compared to SuM-IN connections in the CA2 area? It may be good to elaborate on this a bit in the discussion.

*Reviewer #3:*

In this manuscript, Robert et al., demonstrated that medial SuM send glutamatergic projections to the hippocampal CA2 region, and stimulation of these projections exert mixed excitatory and inhibitory responses in CA2 pyramidal neurons. Furthermore, they showed that SuM-CA2 circuits recruit local PV basket cells to provide feedforward inhibition to CA2 pyramidal cells, which increases the precision of action potential firing in conditions of low and high cholinergic tone. Finally, they performed in vivo electrophysiology recording to show that stimulation of SuM-CA2 projections can influence CA1 activity. Overall, this is a well-designed study, and the quality of the data is high. The authors performed impressive amount of electrophysiology recording in acute slices and provided detailed information on how long-distance SuM projection neurons regulate CA2 pyramidal cell activity. These findings provide insights into how SuM activity directly acts on the local hippocampal circuit to modulate social memory encoding. However, there are some concerns that need to be addressed.

1. The authors performed CAV-based retrograde tracing and demonstrated that medial SuM sends glutamatergic projections to CA2. These results are in contrast to a recent study (Li et al., *eLife* 2020) showing that lateral SuM neurons send dense projections to both CA2 and DG, and the SuM-DG projections release both glutamate and GABA to dentate granule cells. Based on the results from this study and the study from Li et al., does that mean medial SuM neurons are different from lateral SuM neurons in terms of the neurotransmitters they release? The authors need to clarify this point and provide additional ephys data to show that pyramidal cells do not receive direct GABAergic inputs upon stimulation of SuM-CA2 projections using high-chloride internal solution to reveal the IPSCs.

2. The authors claim that SuM-CA2 circuits recruit local PV basket cells to provide feedforward inhibition to CA2 pyramidal cells. While the data presented are supportive, they are not entirely convincing. Specifically, MOR agonist DAMGO is not specific to PV BCs. Though DAMGO has a preferential effect on PV cells over CCK cells, other interneuron types have been shown to be sensitive to DAMGO manipulation. Therefore, these results are subject to alternative interpretation that other types of CA2 local interneurons maybe involved. To show whether PV BCs is the sole interneuron subtype involved, the authors may use a P/Q type calcium channel blocker, ω-agatoxin-TK, as P/Q Ca^2+^ channels are unique to PV BCs. In addition, chemogenetic inhibition of PV BCs was used, but light-evoked IPSCs are not completely blocked. The authors claimed this could be due to partial silencing of PV BCs. However, there is no evidence showing the efficacy of 10µM CNO application in suppressing CA2 PV basket cell activity. These data should be provided in order to draw such conclusion.

3. CCK basket cells are known to excite PV basket cells (Lee et al., 2011) via a pertussin-toxin sensitive pathway. Is it possible that SuM-CA2 mediated excitation of PV basket cells includes a CCK intermediary? This point should be discussed.

4. The in vivo recording data showed that SuM-CA2 circuit stimulation decreases the firing rate of CA1 pyramidal cells followed by increased firing rate in these cells. Then the authors performed slice recording and showed that the reduced firing rate of CA1 neurons in vivo is likely caused by increased inhibitory inputs onto CA1 pyramidal cells. Figure 7G-H seems to explain the reduced events in the first phase of the tetrode recordings, but not the rebound part. Is there some circuit component that is lost when making slices? Furthermore, what does SuM-CA2 circuit stimulation do to theta/γ rhythms in CA1? These data should be available in the tetrode recordings.

---

## [Author Response]

The study describes the properties of inputs from the supramammillary nucleus (SuM) to the CA2 area of the hippocampus. Novel information is presented by the influence of the SuM input on the local hippocampal network in the CA2 and what the effect of this input is on network activity in the CA1. The authors use complimentary methods to address this question including patch-clamp recordings and optogenetics. Overall the reviewer found this study important, the experiments well-designed and the data of high quality. However, there are several key points raised by the reviewers to strengthen the data in order to fully support the authors' conclusions, addressing these will require additional experimental work. The list below summarizes the list of required experiments reviewers agreed would be necessary for having full confidence in the authors' conclusions. Reviewers' individual comments contain additional issues that would require the authors to make changes to the discussion, figure legends etc.1. The authors would need to show the effect of SuM stimulation on on synaptically triggered APs and not only on Aps evoked with a current step.

We agree with the reviewers that establishing the role of SuM input in controlling synaptically-driven AP firing is most relevant. However, unlike other hippocampal pyramidal neurons, studying synaptically-evoked action potential firing in CA2 PNs is very difficult with optimal slice health and without blocking inhibitory transmission since CA2 PNs have low intrinsic excitability and receive strong inhibition as detailed below. Because of their low membrane resistance and very hyperpolarized membrane potential, CA2 PNs have an intrinsically low excitability which makes it difficult for any given input to cause AP firing in basal conditions. Furthermore, stimulation of CA3 axons evokes a very large feedforward inhibition that strongly prevents action potential firing. Likewise, distal dendritic inputs are extremely attenuated and also rarely lead to AP firing under physiological conditions. For these reasons we used current step injection-induced AP firing as a mean to induce spiking and test the effect of SuM input stimulation. Admittedly, this experimental design is somewhat artificial. We have previously discovered that CA2 PNs spontaneously depolarize and fire burst of action potentials under conditions of elevated cholinergic tone. It is known that the properties of these burst are strongly dependent on both excitatory and inhibitory synaptic transmission (Robert et al., 2020), and also the SuM is most likely to be active during theta rhythm, when cholinergic tone is elevated in the hippocampus. Therefore, we examined how the carbachol-induced bursting of CA2 PNs was altered by SuM input stimulation. We think that the results presented in Figure 7 are directly relevant to the reviewers’ request.

We acknowledge that this experimental design does not allow for the contribution of specific inputs to AP bursting to be established. Thus, we performed additional experiments using electrical stimulation of axonal inputs in SR (CA3 axons) or SLM (EC axons) paired with SuM inputs stimulation. This new data is presented in Figure 2 and described in the text (pages 7-8). Unfortunately, for the vast majority of our slices, we were unable to evoke AP firing with reasonable electrical stimulation intensities, as is very typical for CA2 PNs. However, we were able to determine how SuM input summates with SR and SLM synaptic inputs, as well as determine how the summation ratio of these separate inputs is altered by SuM activity. Notably, we observed a reduction in PSP summation from both SR and SLM repeated stimulation when paired with SuM input activation, which is consistent with a shunting effect of the perisomatic-targeting basket cells recruited by SuM to drive inhibition in CA2 PNs.

2. The change in the balance of EPCs and IPSCs in a train should be demonstrated in a single cell.

We thank the reviewers for raising this point which is very important for the interpretation of the data with repeated SuM input stimulation presented in Figures 6, 7 and 8. In supplemental figure 4 we present IPSCs and EPSCs all recorded in the same CA2 PNs in the presence and absence of carbachol (described in pages 14-15). In short, we observed a consistent short-term depression of both SuM-driven monosynaptic EPSCs and di-synaptic IPSCs in CA2 PNs under basal conditions, with the resulting E/I ratio being stable over the train of SuM stimuli. Application of carbachol reduced initial amplitudes of both EPSCs and IPSCs while decreasing their short-term depression, resulting in a decrease of E/I ratio suggesting that the SuM inhibitory influence over CA2 PNs is even stronger and more sustained in conditions of elevated cholinergic tone.

3. The properties of monosynaptic/disynaptic events should be compared and the lack of direct GABAergic input from the SuM demonstrated. The authors should quantify the delay time to light-evoked IPSCs to address whether the SuM-CA2 inputs are forming monosynaptic or disynaptic GABAergic connections to pyramidal neurons, as it is possible SuM neurons co-release glutamate and GABA to CA2. Given the importance of the mono vs. disynaptic innervation of different types of cells, the authors should go beyond the TTX experiments (as TTX would block a disynaptic EPSC) and also use 4-AP to recover the TTX blocked current to unequivocally prove that they inputs are monosynaptic.

The reviewers are completely correct in asking for these experiments in order to confirm that indeed the SuM-CA2 inputs are entirely glutamatergic. All of these points are crucial to our investigation. In fact, we have already performed all of these requested experiments and analysis. These data were published as an extended data figure in Chen et al., *Nature* 2020. The exclusive mono-synaptic glutamatergic nature of the SuM-CA2 transmission was demonstrated and directly compared to the SuM-DG synapse with measurements of EPSC vs IPSC onset latency and application of TTX with 4AP or NBQX + APV. We regret not summarizing more clearly in the text of this manuscript our previous findings and have added a description of these results to our manuscript (page 7).

4. The preferential role of PV+ cells should be shown with a more selective pharmacological approach.

We have taken the advice from reviewer #3 and performed a series of experiments using the P/Q voltage-gated calcium channel blocker ω-agatoxin TK. Block of these channels is predicted to only inhibit Gaba release from PV+ interneurons. These results of these experiments are in figure 5 and described page 12. Our experiments showed a near-complete block of SuM-driven IPSCs in CA2 PNs consistent with PV+ INs mediating this feedforward inhibition. However, the interpretation of these results is complicated by the fact that application of ω-agatoxin TK also reduced glutamatergic transmission at the SuM-CA2 synapse. If we assume that the SuM-IN synapse has a similar population of channels, then part of the reduction in inhibitory transmission may not be due to PV+ IN specificity, but by the reduction in glutamate release. While these results are not fully conclusive proof, we have still included them, as they further characterize the SuM input.

As for the chemogenetic inhibition of PV+ INs, we took the reviewers advice and provided more direct evidence that 10 µM CNO is not sufficiently suppressing CA2 PV basket cell activity. We performed recordings of PV+ INs in area CA2 and measured changes in the intrinsic properties and action potential firing of these cells before and following CNO application. We did observe a small hyperpolarization of the membrane potential, however we still observed AP firing in several of the cells, indicating that this hyperpolarization is insufficient to fully silence PV+ INs. This is shown in supplemental figure 3 and described on page 11. GiDREADD can also act on pre-synaptic terminals, so it is not entirely conclusive if the PV+ cells that fire action potentials still release GABA. However, we feel that in combination with the incomplete expression of the GiDREADD in all PV+ interneurons, and the residual AP firing, that the 10 µM CNO is not entirely effective at suppressing PV+ IN activity in our experiments.

We would like to mention that the expression of DOR receptors has been shown to be expressed in 35% of PV+ BCs in the hippocampus (Erbs, 2012) and furthermore, we have shown that transmission exclusively from PV+ cells in area CA2 undergoes a DOR-dependent depression (Piskorowski and Chevaleyre, 2013) that can be induced by application of DPDPE. We conclude that the effect of DPDPE on SuM transmission is compelling evidence that these inputs target PV+ cells.

5. The authors should elaborate on how SuM stimulation influences theta/γ rhythms in the CA1 area.

We have added a section to the discussion (pages 19-21) about theta/γ rhythms and the potential contribution of SuM input. We agree whole-heartedly with the reviewer about the fundamental importance of examining how SuM contributes to these brain activities in vivo. Please see our extended answer to reviewer 3 below.

Reviewer #1:In this study Robert et al. describes the properties of long-range projections from the SuM to the CA2 area of the hippocampus. The authors identified direct excitatory and indirect inhibitory drive from SuM inputs on CA2 pyramidal neurons and showed that direct excitatory drive impinges on PV-positive basket cells. The overall effect of the input on CA2 activity was an increased precision of APs. The study also suggests that the input from the CA2 drives inhibition in the CA1 area. The study provides very interesting and new information about the cellular properties of SuM input in the CA2 area. This is an important question given the increasing importance of SuM inputs in social memory encoding. The study is timely, currently we have very limited data about the features and exact cellular profile of this input. The study is using elegant technical approaches o answer the central question of the study. While the study is addressing an important question and provides novel data, the authors central claim about the role of feedforward inhibition would need to be strengthened by the addition of experiments addressing how E-I balance changes in trains in individual neurons and how this can be linked to changes in the temporal precision of synaptically evoked APs.Action potentials are evoked with a current step. Since the study is focused on the network effects of feedforward inhibition, it would be useful to see how the properties of synaptically evoked action potentials change. In the cortex and in the CA1 feed forward inhibition was shown to limit the temporal summation of excitatory inputs which lead to decrease in AP jitter (Gabernet et al., 2005, Pouille and Scanziani, 2001). In order to map these dynamics APs should be evoked via synaptic stimulation and not through current injection.

Please see our response to point #1 in the above section.

The authors show recordings of monosynaptic EPSCs in pyramidal cells and interneurons. It would be important to know how inhibitory and excitatory PSCs change in a train. Recordings from single cells held at E-GLUT ad E-GABA would allow the authors to monitor excitatory and inhibitory events in a train ad map how their balance changes. Can the change in E-I balance explain the change in AP jitter?

These experiments examining the changes in E/I balance with a train of inputs has been added to the manuscript (Supplemental figure 4, pages 14-15). We thank the reviewer for this insight. In fact, the E-I balance does not change during a train of stimulation, staying between 0.5 and 0.6 (supplemental figure 4). However, we do see a large change in the E/I ratio in high cholinergic tone (that also does not change during the train), indicating that the inhibitory drive of the SuM input is increased under these conditions.

What are the characteristics of the SuM-driven inhibitory currents? Does the latency and jitter of monosynaptic EPSCs and disynaptic IPSCs differ? If one is monosynaptic and the other is disynaptic one would expect significant differences in both of these parameters.

These points are very critical to understanding the local SuM-CA2 circuitry. These experiments have been published as supplemental data for Chen et al., Nature 2020. We have also added a description of these results to our current manuscript (page 7).

How do the authors exclude the contribution of feed-back inhibition? feedforward and feed-back inhibition both could have an impact on the temporal precision of APs.

We thank the reviewer for raising this important point which we failed to address in our initial submission. In basal conditions, feedback inhibition is not at play because stimulation of the SuM input evokes only small amplitude EPSPs that remain subthreshold in CA2 PNs, as documented in Figures 1 and 2. Indeed, we never observed SuM-evoked AP firing from CA2 PNs in our recordings (n = 0 out of 78, Figure 3). However, when inducing action potential firing in CA2 PNs either by current injection (Figure 6) or CCh application (Figure 7), the contribution of feedback inhibition becomes very relevant. Indeed, it is likely that CA2 PNs recruit not only feedback inhibition (Mercer, 2012a, Mercer, 2012b) but also recurrent excitation (Cui, 2013, Hitti, 2014, Okamoto, 2019), both of which could impact the temporal precision of APs. In the case of current step injection-induced firing, the contribution of feedback inhibition and recurrent excitation to reducing AP jitter can be examined with our pharmacology experiments: in the GABA receptors blocked condition (Figure 6), the effect of SuM input stimulation on both the AP latency and jitter is abolished, thereby indicating that the main factor imposing the initial AP delay and constraining AP jitter is SuM-driven feedforward inhibition. This is because, at least for the first AP evoked by current step injection in the single CA2 PN recorded from, it is reasonable to assume that other CA2 PNs were not actively firing thus removing the possibility of feedback inhibition and recurrent excitation from the equation. Furthermore, the small probability of PNs to be connected (1.4%) and the small size of CA2-CA2 EPSP (~0.5mV) (Okamoto, 2019) makes it very unlikely that recurrent excitation plays any role when a single PN is depolarized. The problem becomes much more complicated when considering the CCh-induced AP burst firing conditions, as many CA2 PNs likely burst fire somewhat independently of the single CA2 PN recorded in any given experiment. Therefore, feedback inhibition and recurrent excitation are to be expected in these conditions, thereby potentially contributing to the temporal precision of AP firing in the recorded CA2 PN. Unfortunately, typical pharmacology blocking GABA or glutamate receptors is not suitable here as the CCh-induced AP burst firing pattern of CA2 PNs relies on intact synaptic transmission (Robert, 2020), thus making the examination of the respective roles of feedforward versus feedback inhibition in these conditions difficult to probe experimentally. Nevertheless, we deem reasonable to interpret our data with SuM input stimulation during CCh-induced spontaneous AP burst firing as follows: by recruiting perisomatic-targeted feedforward inhibition on most CA2 PNs, SuM input stimulation might effectively enforce a restricted time window during which CA2 PNs can undergo or sustain AP burst firing, thereby limiting independent burst firing from individual CA2 PNs and instead shaping a synchronized output from many CA2 PNs subsequently driving feedforward inhibition over area CA1 as shown in Figures 7 and 8.

Reviewer #2:The article brings to light the functional consequences of the activity of SuM afferents terminating at CA2 neurons in the hippocampus using a combination of a variety of methods like whole-cell voltage clamp and optogenetics. In addition, the authors provide evidence that modulation of the CA2 neurons by SuM afferents affects the activity pattern of CA1 neurons. Specifically, the study reveals that the 'functional' connectivity between SuM and CA2 is mainly mediated by the regulation of PV+ basket cells that are involved in the feed forward inhibition of CA2 principal neurons. This study is also relevant in the context of neuropsychiatric disorders where PV+ IN density in the CA2 area is preferentially reduced.It would be good if some results and implications are further clarified for better understanding in the Discussion section:1. The results indicate that SuM recruits a feed forward inhibition onto CA2 PNs, which contributes to the shaping of CA2 AP firing. However, it is not entirely intuitive how the feed forward inhibition of CA2 PNs by SuM also reduces CA1 activity, as CA2 has also been known to recruit strong feed forward inhibition onto CA1. This would intuitively suggest that decrease in CA2 activity by photostimulation of SuM afferents will in turn decrease the feed forward inhibition by CA2 onto CA1, and thereby increase CA1 activity. However, the results suggest otherwise. Would this be suggestive of a stronger direct excitatory projection from CA2 to CA1 PNs that is more dominant than the feed forward inhibition of CA1 PNs by CA2? This may be a good point to further elaborate on in the Discussion section, so that the effect of SuM-CA2 connectivity on CA1 output becomes clearer.

The referee is correct in that the connection between CA2 and CA1 is predominantly excitatory. CA2 PNs make a powerful excitatory connection with CA1 PNs (Chevaleyre and Siegelbaum, 2010, Kohara et al., 2014). CA2 PNs also drive feedforward inhibition onto CA1 PNs (Boehringer et al., 2017, Nasrallah et al., 2019). Here we show that under conditions that resemble active exploration (high cholinergic tone) the SuM-CA2-CA1 elicits a reliable poly-synaptic IPSC in CA1 PNs (Figure 8), as well as an occasional poly-synaptic CA1 EPSC. Our data does not directly show that SuM feedforward inhibition of CA2 causes a drop in CA2-CA1 excitation underlying the decreased CA1 activity following SuM-CA2 stimulation. Instead, our data shows that SuM inhibition of CA2 yields synchronized feedforward inhibition from CA2 to CA1 which decreases CA1 activity. We appreciate the complexity of this idea and have done our best to elaborate in the discussion (page 19).

2. In the introduction section line 44, it is written that 'CA2 neurons do not undergo NMDA-mediated synaptic plasticity'. This may not be always the case; rather it may be better to rephrase 'NMDA-mediated' as 'high frequency stimulation-induced'. It has been shown previously that NK1 receptor activation by pharmacological application of substance P in hippocampal slices triggers a slow onset NMDA-dependent LTP in CA2 neurons by high frequency stimulation of CA3 afferents to CA2 (Dasgupta et al., 2017).

We thank the reviewer for this clarification. In the text, “NMDA-mediated” has been replaced by “high frequency stimulation-induced” as suggested.

3. Line 250: "BC transmission is insensitive to MOR activation (Glickfeld et al., 2008)."Was the Glickfeld study done in CA2 neurons? If not, it would be good to show that PV+ CA2 BCs are also sensitive to DAMGO and to what degree?The experiment shows that IPSC in PNs are inhibited by DAMGO that should have enhanced light induced EPSCs if PV+ BCs are responsible for feed forward inhibition. But it seems that has not been observed. What are direct EPSCs – electrical stimulation of CA3-CA2 synapses?

In the experiments where we applied DAMGO, the EPSCs were recorded in presence of GABAR blockers. Therefore, any change in GABA transmission would not affect the light induced EPSC.

The Glickfeld study was performed in CA1 only. However, expression of MOR mRNA is very strong in area CA2 and a large fraction of PV+ INs are also positive for MOR throughout the hippocampus (Ralf et al., Journal of Comparative Neurology, 2004). We haven’t directly monitored the effect of DAMGO on PV+ transmission in area CA2. However, in an ongoing study, we stimulated CA3 inputs and monitors the electrically evoked IPSCs in CA2 pyramidal cells (Author response image 1). We found that DAMGO application strongly reduces IPSCs (by ~75%) and completely occludes subsequent effect of the DOR agonist DPDPE. We are confident that DPDPE is acting solely on PV+ cells in area CA2. Therefore, from these results we can conclude that PV+ INs in area CA2 are also strongly sensitive to DAMGO as they are known to be in area CA1.

4. Overall, the results seem to suggest that SuM stimulation would induce a net inhibition (?) of CA2 PNs by recruiting interneurons (INs). However, the role played by the direct glutamatergic connections from SuM to CA2 PNs is not entirely clear. Is it less prominent due to sparse SuM-PN projections compared to SuM-IN connections in the CA2 area? It may be good to elaborate on this a bit in the discussion.

We feel that the combined excitation and feedforward inhibition from the SuM to area CA2 is acting together to coordinate the timing of CA2 PN output. We have expanded this explanation in the discussion (page 18).

Reviewer #3:In this manuscript, Robert et al. demonstrated that medial SuM send glutamatergic projections to the hippocampal CA2 region, and stimulation of these projections exert mixed excitatory and inhibitory responses in CA2 pyramidal neurons. Furthermore, they showed that SuM-CA2 circuits recruit local PV basket cells to provide feedforward inhibition to CA2 pyramidal cells, which increases the precision of action potential firing in conditions of low and high cholinergic tone. Finally, they performed in vivo electrophysiology recording to show that stimulation of SuM-CA2 projections can influence CA1 activity. Overall, this is a well-designed study, and the quality of the data is high. The authors performed impressive amount of electrophysiology recording in acute slices and provided detailed information on how long-distance SuM projection neurons regulate CA2 pyramidal cell activity. These findings provide insights into how SuM activity directly acts on the local hippocampal circuit to modulate social memory encoding. However, there are some concerns that need to be addressed.1. The authors performed CAV-based retrograde tracing and demonstrated that medial SuM sends glutamatergic projections to CA2. These results are in contrast to a recent study (Li et al., eLife 2020) showing that lateral SuM neurons send dense projections to both CA2 and DG, and the SuM-DG projections release both glutamate and GABA to dentate granule cells. Based on the results from this study and the study from Li et al., does that mean medial SuM neurons are different from lateral SuM neurons in terms of the neurotransmitters they release? The authors need to clarify this point and provide additional ephys data to show that pyramidal cells do not receive direct GABAergic inputs upon stimulation of SuM-CA2 projections using high-chloride internal solution to reveal the IPSCs.

This is a very valid point, and as we have detailed above, the experiments directly comparing the SuM transmission into the DG and area CA2 have been published as a supplemental figure in Chen et al., *Nature* 2020. As the reviewer mentions, in that publication we present multiple lines of evidence supporting the premise that the CA2-projecting and DG-projecting cells in the SuM are from different populations and use different neurotransmitter modalities. These results are supported by prior histological results by Soussi et al., EJN, 2010 that show the CA2-projecting SuMM fibers do not co-express GAD65 unlike the DG-projecting SUML fibers that express both VGLUT2 and GAD65.

In the Li et al., paper (*eLife*, 2020) the authors show that AAV5-DiO.eYFP injected into the SuM in a Vgat-cre mouse line results in fibers in area CA2. These results lack control data showing the specificity of the AAV5 viral vector in the vgat-cre line (co-labelling with Gad65/67…). In our experience, it is very easy to get non-specific expression of AA5 vectors in area CA2. In fact, we use this problem to our advantage in Dominguez et al., *Cell Reports* 2019.

2. The authors claim that SuM-CA2 circuits recruit local PV basket cells to provide feedforward inhibition to CA2 pyramidal cells. While the data presented are supportive, they are not entirely convincing. Specifically, MOR agonist DAMGO is not specific to PV BCs. Though DAMGO has a preferential effect on PV cells over CCK cells, other interneuron types have been shown to be sensitive to DAMGO manipulation. Therefore, these results are subject to alternative interpretation that other types of CA2 local interneurons maybe involved. To show whether PV BCs is the sole interneuron subtype involved, the authors may use a P/Q type calcium channel blocker, ω-agatoxin-TK, as P/Q Ca^2+^ channels are unique to PV BCs. In addition, chemogenetic inhibition of PV BCs was used, but light-evoked IPSCs are not completely blocked. The authors claimed this could be due to partial silencing of PV BCs. However, there is no evidence showing the efficacy of 10µM CNO application in suppressing CA2 PV basket cell activity. These data should be provided in order to draw such conclusion.

As the reviewer suggested, we used ω-agatoxin-TK to explore whether PV+ BCs are the sole population of interneurons. Consistent with PV+ cells being the primary population underlying the feedforward inhibition onto CA2 PNs, we saw a nearly complete block of light-evoked IPSCs. However, we also examined the direct excitatory SuM-CA2 glutamatergic synapse and found that application of ω-agatoxin-TK resulted in a 50% block of glutamatergic transmission. Thus, if we assume that the glutamatergic SuM-IN synapse is also reduced then the reduction of feedforward IPSC may not be because of the specificity of P/Q Ca^2+^ channel expression of PV+ cells, but because the SuM glutamatergic transmission is reduced. While these results do not provide the definitive result we were hoping for, we have still included them in Figure 5 to be considered in parallel with the other experiments (pages 11-12).

As for the chemogenetic inhibition of PV BCs, we took the reviewers advice and provided more direct evidence that 10 µM CNO is not sufficiently suppressing CA2 PV basket cell activity. We performed recordings of PV+ INs in area CA2 and measured changes in the intrinsic properties and action potential firing of these cells before and following CNO application. We did observe a small hyperpolarization of the membrane potential, however in we still observed AP firing in several of the cells, indicating that this hyperpolarization is insufficient to fully silence PV+ INs. This is shown in supplemental figure 3 and described page 11. GiDREADD can also act on pre-synaptic terminals, so it is not entirely conclusive if the PV cells that fire action potentials still release GABA. However, we feel that in combination with the incomplete expression of the GiDREADD in all PV+ interneurons, and the residual AP firing, that the 10 µM CNO is not entirely effective at suppressing PV-BC activity in our experiments.

3. CCK basket cells are known to excite PV basket cells (Lee et al., 2011) via a pertussin-toxin sensitive pathway. Is it possible that SuM-CA2 mediated excitation of PV basket cells includes a CCK intermediary? This point should be discussed.

This is a very intriguing point, and we thank the reviewer for this suggestion. Lee et al. 2011 demonstrates that CCK application results in a depolarization of PV+ cells via a nonselective calcium conductance linked to intracellular calcium store release. The idea that CCK release would potentially enhance ability of SuM inputs to recruit PV+ cells is quite interesting and have added this to the discussion (page 19). Similarly, we have also mentioned the other intriguing finding that CCK cells actively inhibit PV+ basket cells (Dodek 2021), presenting two ways in which CCK cells may be acting indirectly to modulate SuM input to area CA2 (page 19).

4. The in vivo recording data showed that SuM-CA2 circuit stimulation decreases the firing rate of CA1 pyramidal cells followed by increased firing rate in these cells. Then the authors performed slice recording and showed that the reduced firing rate of CA1 neurons in vivo is likely caused by increased inhibitory inputs onto CA1 pyramidal cells. Figure 7G-H seems to explain the reduced events in the first phase of the tetrode recordings, but not the rebound part. Is there some circuit component that is lost when making slices? Furthermore, what does SuM-CA2 circuit stimulation do to theta/γ rhythms in CA1? These data should be available in the tetrode recordings.

This is a very interesting point, and a central flaw in acute slice physiology. There are definitely circuit components that are lost when making transverse hippocampal slices. We have added two paragraphs to the Discussion section detailing the potential role of SuM-CA2 input to theta and γ oscillations (pages 19-21).

The in vivo data in the paper was based on short stimulation pulses in resting animals where theta and γ are minimal. Thus, further examination of theta and γ requires further distinct experiments. As this manuscript elucidates the local circuitry underlying the SuM input in area CA2, and the overall effect of SuM activity on action potential firing in this region, further examination of network oscillations is outside the scope of this work.

However, we still went through all of our data to try to determine if there is a large apparent effect of SuM activity in theta / γ. Because of the current Covid-19 crisis, we have not been able to continue with the majority of our experimental work for this project, so we can only present preliminary data from two animals with acceptable trials, tetrode locations (in CA1), fiber placement over the area CA2, and properly targeted virus in the SuM. We present in Author response image 2 our very preliminary results of CA1 LFP with and without optical stimulation on a circular linear track. As the mice ran on the track, we applied a protocol with 1 minute ON, 1 minute OFF of 10 Hz light stim in CA2. For analysis, we calculated the PSD from the CA1 LFP with the middle 40” of all laser ON periods and the middle 40” of all laser OFF periods, then averaged. Power across the theta (4-12Hz) and γ (30-90 Hz) bands was normalized to the total power in the 0-4 Hz band for each trial, then averaged across the two animals.

**Author response image 2. respfig2:** 

As visible in the wavelet spectragrams across the whole trial, we do not see any obvious change in theta or γ power when the laser is on. We also present (Author response image 3) the normalized power in the theta and γ bands in OFF/ON conditions and averaged across the 2 mice. Currently we do not see any obvious changes with this preliminary experimental design. These results have convinced us that this question is not easily answered in these challenging times and is needs to be explored in a later study.

**Author response image 3. respfig3:**